low-temperature physics/solid-state physics/thermodynamics

specific heat, field theory, sound velocity, boson peak, dimensionless thermodynamics, glasses

**Author for correspondence:**
Yuri Vladimirovich Gusev
e-mail: yuri.v.gusev@gmail.com

# The quasi-low temperature behaviour of specific heat

## Yuri Vladimirovich Gusev[1,2]

[1]Lebedev Research Center in Physics, Leninsky Prospekt 53, str. 11 (38), Moscow 119991, Russia
[2]Department of Physics, Simon Fraser University, 8888 University Drive, Burnaby, BC, Canada V5A 1S6

 YVG, 0000-0001-8246-9969

A new mathematical approach to condensed matter physics, based on the finite temperature field theory, was recently proposed. The field theory is a scale-free formalism; thus, it denies absolute values of thermodynamic temperature and uses dimensionless thermal variables, which are obtained with the group velocities of sound and the interatomic distance. This formalism was previously applied to the specific heat of condensed matter and predicted its fourth power of temperature behaviour at sufficiently low temperatures, which was tested by experimental data for diamond lattice materials. The range of temperatures with the quartic law varies for different materials; therefore, it is called the quasi-low temperature regime. The quasi-low temperature behaviour of specific heat is verified here with experimental data for the fcc lattice materials, silver chloride and lithium iodide. The conjecture that the fourth order behaviour is universal for all condensed matter systems has also supported the data for glassy matter: vitreous silica. This law is long known to hold for the bcc solid helium-4. The characteristic temperatures of the threshold of the quasi-low temperature regime are found for the studied materials. The scaling in the specific heat of condensed matter is expressed by the dimensionless parameter, which is explored with the data for several glasses. The explanation of the correlation of the 'boson peak' temperature with the shear velocity is proposed. The critique of the Debye theory of specific heat and the Born−von Karman model of the lattice dynamics is given.

# 1. The field theory for condensed matter physics

## 1.1. The scaling in thermal theory

Perhaps, the most powerful and deep idea of theoretical physics is *the scaling*, e.g. [1]. This concept is used throughout physical sciences and specifically in thermodynamics [2] and condensed

matter physics [3]. The scaling embodies the very essence of a physical theory, that a physical law, expressed as a mathematical equation, can describe different instances of the same physical phenomenon, only if appropriate physical variables are chosen.

Thermal phenomena delivered one of the first examples of the scaling in physics. The Dulong–Petit rule [4,5] was discovered when these scientists realized that the specific heat of chemical elements in solid form should be measured per unit of mole, i.e. per number of atoms, not per unit of mass. The scaling appeared in the theory of heat capacity of P. Debye [5,6], who developed the idea of heat as the energy density of standing sound waves in condensed matter. The Debye function is an integral over the exponential function, which necessarily has a dimensionless variable. The dimensionless property was achieved by using a characteristic temperature called the Debye temperature, which strips the physical dimensionality of thermodynamic temperature. There is only one characteristic temperature in the Debye theory because it assumes a single characteristic velocity of sound, averaged over the longitudinal and transverse velocities in condensed matter. The calculation of this temperature from the elastic properties [7] and its comparison with the one derived from the specific heat data had much attention in the past. However, phenomenologically, the Debye theory needs at least three characteristic parameters for different temperature regimes [8]. The Debye model is still used now, even though it is theoretically inconsistent [9] (see more detail in §5.1) and it cannot correctly describe the specific heat behaviour close to the critical points or at low temperature (which may not be near absolute zero)—the subject of this paper.

The ultimate form of the scaling is the scale-free (conformal) theory. For example, the field theory serves as a mathematical foundation for the modern physics of elementary particles [10]. The field theory showed us a path to the axiomatic building of physical theories. Following this path, we recently suggested to use the method of the evolution kernel [11,12] (traditionally but erroneously called 'the heat kernel') as the basis for the finite temperature field theory [13]. The finite temperature field theory possesses continuous variables and does not deal with discrete constituents of matter. Therefore, it resembles the thermodynamics of Gibbs ensembles and differs from the statistical mechanics of Maxwell. The finite temperature field theory was then employed to make a proposal of the field theory of specific heat [9]. This model led to some interesting findings about condensed matter physics. It matched well with the data of the specific heat of single crystals of silicon and germanium at low temperatures [9]. This theory will be fully calibrated in a forthcoming work, which will also contain the confirmation of predictions made in [9] for two other elements of the carbon group, natural diamond and grey tin.

In this paper, we deal only with the specific heat of crystalline and non-crystalline matter at 'low temperature'. We demonstrate that the obtained thermal sum may be universally applicable to condensed matter systems by testing it with the data for face-centred cubic crystals and glasses. Within the finite temperature field theory, the notion of 'low temperature' does not exist, because the *absolute* scale of temperature is foreign to a scale-free theory. This fact brings up the requirement of a dimensionless variable. Discarding the absolute scale of thermodynamic temperature forces us to suggest the term 'quasi-low temperature' (QLT) regime, which depends on material properties and can range from a few kelvin to almost 200 K [9].

Let us proceed with an overview of theoretical ideas and mathematical expressions. We recall basic equations of the field theory of specific heat while referring for the details to [9,13]. The field theory formalism is developed in the four-dimensional *Euclidean* space, i.e. there is no physical time selected by the Minkowski (Lorentzian) signature of the metric, i.e. this is not relativistic set-up, even though the total dimension of space–time is *four*. Time is still singled out by its closed topology, $\mathbb{R}^3 \times \mathbb{S}^1$. This notation means that the physical space–time presents a three-dimensional spatial domain, $\mathbb{R}^3$, which plays a role of a material body, with the one-dimensional closed manifold, $\mathbb{S}^1$, which plays a role of the imaginary (due to the Euclidean metric signature) and periodic physical time. These mathematical derivations belong to *geometry* [13] and should not necessarily refer to temperature (or energy). By a well established in theoretical physics conjecture, thermodynamic temperature, $T$, is inversely proportional to the orbit's length of the closed coordinate of $\mathbb{S}^1$, which we called the Planck's inverse temperature, $\beta$. This variable is used in the finite temperature quantum field theory [13]. It is common in theories for the heavy-ion collision experiments [14] and expressed as,

$$\beta = \frac{\hbar c}{B k_B T},$$

(1.1)

where $\hbar$ is the Planck constant, $k_B$ is the Boltzmann constant, $c$ is the speed of sound and $B$ is the experimental calibration constant. The principal idea of [9] is that the same mathematics can be

applicable to various physical theories, where instead of the speed of light, a characteristic velocity of other physical phenomena, $v$, can be used. In general, the Planck's inverse temperature, $\beta$, is a better thermal variable than the thermodynamic temperature, $T$, because it does not allow the absolute zero of temperature. It also modifies the thermal scale in such a way that it stretches the region of low temperatures, where the wealth of condensed matter physics occurs and shrinks the less physically interesting region of high temperatures.

## 1.2. The thermal variables and the thermal sum

In a physical theory, one can tell whether a value of a physical quantity is small or large when it is compared with some reference value of the same quantity, the act of comparison being called the measurement. The use of the thermodynamic temperature in thermal physics implies the existence of an absolute value of temperature, which is universal for all materials and for any physical conditions. By convention, the absolute zero of temperature is used as this reference value; however, this is a wrong choice for two reasons. First, in classic thermodynamics, the absolute zero of temperature cannot be reached, thus, it effectively does not exist. However, there may exist a limit, $T/T_{ref} \to 0$, with respect to some finite reference temperature, $T_{ref}$. Only this asymptotics is mathematically correctly defined because $T/T_{ref}$ is dimensionless. Second, any absolute value of temperature is useless as a reference temperature because thermal properties of materials vary so vastly, e.g. the triple point of water is (exactly) 273.16 K, while the triple point of equilibrium hydrogen is 13.81 K [15, p. 167]. Therefore, experimental evidence and mathematical consistency force us to seek a suitable thermal variable instead of a fixed reference temperature.

The use of dimensionless variables in the field theory, whose functionals are dimensionless by construction, is mathematically justified. In general, only expansions in the powers of a dimensionless variable are mathematically consistent and, therefore, physically sound. It is natural to see how the field theory derivations [9] produce the dimensionless variable,

$$\alpha = \frac{1}{B}\frac{\hbar}{k_B}\frac{v}{aT},$$  (1.2)

where $a$ is the average interatomic distance, $v$ is the velocity of sound and $B$ is the experimental calibration constant. The average interatomic distance $a$ serves two purposes. Not only does it make the variable $\alpha$ dimensionless, but it also quantifies the limit of the validity of the approximation of continuous medium for condensed matter, which consists of discrete constituents (atoms). The variable (1.2) was declared a proper variable for the thermal physics of condensed matter (for crystalline matter, there are several lattice constants along crystallographic directions), and it replaced the thermodynamic temperature $T$ (kelvin).

The use of the definition (1.2) in place of the temperature assumes the constancy of the ratio of the sound velocity and the lattice constant (or the average interatomic distance in disordered matter), $v/a$. Because sound velocities change with temperature, as do lattice constants due to thermal expansion (at constant pressure), this question requires a special consideration. The variation of sound velocities with temperature for the diamond lattice materials (diamond, germanium) was found to be acceptably small [9, p. 60]. Beside the diamond lattice type elements, the same reference [16] gives a graph of the relative change of the sound velocity in $\alpha$-quartz (crystalline silica): about $2 \times 10^{-5}\,\mathrm{K}^{-1}$. Temperature dependence of the sound velocities in silicate glasses are given as graphs in [17]. The velocity of sound, over the whole temperature range of 1100 K, changes noticeably, and the total relative change for the transverse velocity at the frequency 1 THz is about 6%, or the rate of $6 \times 10^{-5}\,\mathrm{K}^{-1}$. While, in general, it would not be acceptable to take $v$ as a constant, it can be considered as such, in the first approximation, with regard to the large changes of absolute values of $T$ in (1.2). We remark that the terahertz frequencies used in the work [17] are the frequencies of thermal energy, because, due to the lattice constant cut-off $a$, no higher frequencies of sound can be supported within condensed matter.

The temperature behaviour of the linear coefficients of thermal expansion, which define the temperature change of interatomic distances also contributes to the variability of $v/a$. However, here it is sufficient to consider only general properties of this phenomenon by estimating its order of magnitude. The typical relative variation of $a$ due to thermal expansion is about $10^{-6}\,\mathrm{K}^{-1}$ [18]. For crystals considered in the following sections, this coefficient is about $10^{-5}\,\mathrm{K}^{-1}$ for temperatures above 100 K [18]; at cryogenic temperatures, it is usually even less. The measurements of vitreous silica at $T$ from 4 to 300 K give the thermal expansion coefficient of about $10^{-6}\,\mathrm{K}^{-1}$. It is clear then that thermal

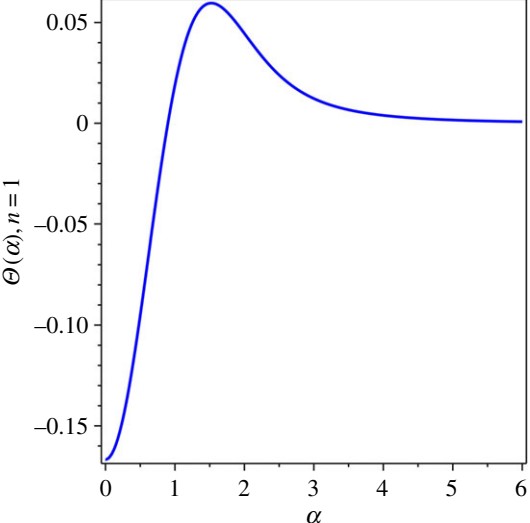

**Figure 1.** $n = 1$ term of $\Theta(\alpha)$.

expansion can be neglected, because even for the whole range of measured temperatures, about $10^3$ K, it can introduce in total the relative change of $a$ of the order of $10^{-3}$. In fact, the temperature dependences of sound velocities and thermal expansion (the size of a body, not the linear coefficient of thermal expansion) of many condensed matter systems correlate. While the sound velocity decreases with the decreasing temperature, e.g. [17], so does the interatomic distance because the size of a matter body decreases (it shrinks) [19]. There are, of course, some anomalies for certain materials (contraction—negative expansion), and thermal expansion is anisotropic in crystalline matter, but these features could be studied and incorporated into the theory later.

In the finite temperature field theory [13], using the evolution kernel in three dimensions and one closed Euclidean time, we obtained the universal thermal functional,

$$-F(\alpha) \propto \sum_{n=1}^{\infty} \frac{1}{n^4 \alpha^3} (1 - \exp(-\alpha^2 n^2) - n^2 \alpha^2 \exp(-\alpha^2 n^2)), \tag{1.3}$$

where unessential numerical factors are discarded. In the high energy theory, similar functionals are often called 'free energy', but this name is misleading and historical, since $F(\alpha)$ is not the free energy of thermodynamics, but rather a dimensionless functional of the field theory. In the sum (1.3), $n$ counts the number of windings of the world function in the closed Euclidean time [13], which is reminiscent of the ensembles of thermodynamics of J.W. Gibbs [20].

Similar ideas were employed in the formalism of the thermal Green functions [14]. The crucial difference is that the sum (1.4) has no zero term, $n = 1, \ldots \infty$, because the very definition of $F(\alpha)$ as well as the effective action [21] requires at least one closed loop. The correction of this error, which was the legacy of the phase space formalism, removed non-existing divergences from physical quantities. The regularization and renormalization of divergent quantities is done in the quantum field theory [14,22]. Here we deal with mathematical expressions that belong to the geometrical analysis and do not use the quantum field theory in its traditional form. The result is a phenomenological physical theory free of the ambiguities of a quantum theory.

Next, the dimensionless thermal sum $\Theta(\alpha)$ was calculated as the derivative of the thermal functional (1.3),

$$\Theta(\alpha) = \sum_{n=1}^{\infty} \frac{1}{n^4 \alpha^4} \{1 - \exp(-\alpha^2 n^2) - n^2 \alpha^2 \exp(-\alpha^2 n^2) - \frac{2}{3} n^4 \alpha^4 \exp(-\alpha^2 n^2)\}. \tag{1.4}$$

This expression was proposed to serve as the thermal function that reflects the variation of thermal energy of a condensed matter system with respect to the physical variables in (1.2). Let us look at the function (1.4) in plots. First, a single mode of the thermal sum, $\Theta(\alpha)$ for $n = 1$, reveals its main mathematical properties in figure 1. The plot of the thermal sum (1.4) can be done by the numerical evaluation of the finite sum, $n = 1 \ldots n$ ($n = 10\,000$ and the vertical axis at $\alpha = 0.01$ in figure 2). The single term is finite negative at $\alpha = 0$, but the whole sum (1.4) diverges as $\alpha \to 0$. Its limit at $\alpha = 0$

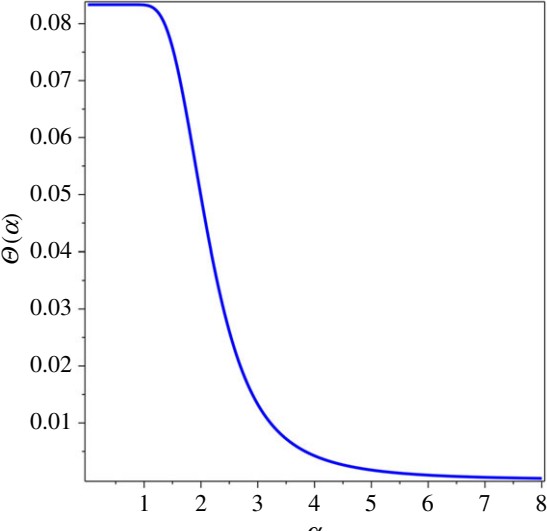

**Figure 2.** $\Theta(\alpha)$ for $n = 10\,000$.

does not exist (1.2), but it can be approached infinitesimally close when $n \to \infty$. In fact, the sum converges rather quickly, and for practical purposes as few as $n = 1000$ terms can be sufficient.

The function $\Theta(\alpha)$ is a mathematical (dimensionless) function. The definition of the variable $\alpha$ connects it to physics. When the molar specific heat of a condensed matter system was derived, the *observable* (in this case, specific heat) is, due to its physical nature, expressed via physical (dimensionful) constants. In the field theory formalism, this is done by the multiplication of (1.4) with proper physical constants, i.e. the gas constant, $R = k_B N_A$. This operation was introduced *axiomatically*, because one cannot obtain physical (dimensionful) quantities from geometrical (dimensionless) functionals in any other way. The standard way to derive physical quantities in quantum field theory was to pass from the field formalism to the *particle* formalism. Particles (or quasiparticles) of the latter appear as quanta of the quantized field, e.g. phonons are the quanta of acoustic waves. Particles possess energy and may possess mass, thereby, observables in the particle formalism are dimensionful by derivation. In the phenomenological theory we develop, the field (acoustic wave) is not quantized. Furthermore, the quantization is not even defined in this theory, which employs the operator methods in differential geometry [12,23]. Because the physical constants of thermal physics, e.g. the gas constant, were established within the statistical thermodynamics of *gases* [24], the constants of the proposed thermal theory of condensed matter are different. Therefore, the field theory of specific heat should be *calibrated* by experimental data in order to determine its two constants, $A$ and $B$, which are unknown numerical coefficients. By conjecture, these constants are the same for a certain class of materials. The following expression is a contribution to the molar specific heat, in units $\mathrm{J\,K^{-1}\,mol^{-1}}$, from one velocity of sound,

$$C_m = A k_B N_A \Theta(\alpha). \tag{1.5}$$

Here $A$ is a calibration constant, which is a dimensionless quantity, i.e. a pure number. It is different from the one defined in [9] because the number of atoms per unit cell and unessential numerical coefficients are absorbed into $A$ of (1.5). It is assumed that all independent velocities of sound (longitudinal and transverse ones, with all their degeneracies) contribute to the observed specific heat.

When plotted as a function of the inverse variable $\kappa = 1/\alpha$, the universal function $\Theta(1/\kappa)$ resembles the behaviour of the specific heat of solid matter, figure 3. Even though it is a contribution from a single velocity, the graph in figure 3 is qualitatively similar to typical graphs of the specific heat data published in the literature. This comparison is quantified and confirmed below. The full calibration of the proposed theory, up to the critical point (melting or ablation, called here the Dulong–Petit limit), cannot be done yet without resolving several technical problems (thermal expansion at pre-melting temperature and several variables $\alpha_i$ corresponding to several velocities of sound). However, the asymptotics of specific heat at the quasi-low temperature can still be studied.

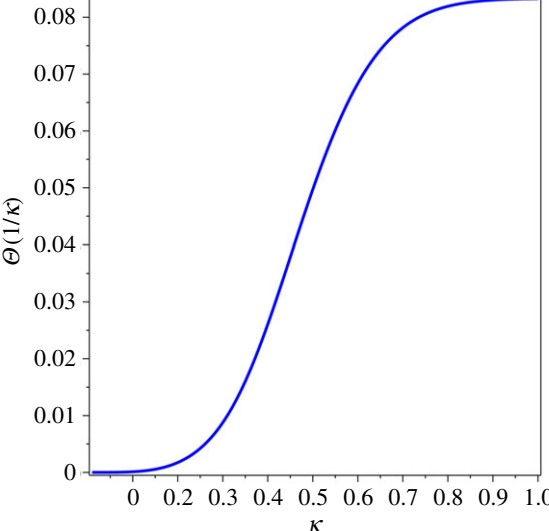

**Figure 3.** $\Theta(1/\kappa)$ for $n = 10\,000$.

## 1.3. The quasi-low temperature behaviour of the specific heat

The quasi-low temperature behaviour of the specific heat (1.5) is determined by the $\alpha \to \infty$ asymptotics of the function $\Theta(\alpha)$,

$$C_m^{(i)} = A_i k_B N_A \frac{\pi^4}{90} \frac{1}{\alpha_i^4}, \quad \alpha_i \to \infty. \tag{1.6}$$

This is equation (1.5) with the index $i$, which denotes a contribution from the corresponding velocity of sound supported within a condensed matter system, including degeneracies, when different velocities in anisotropic matter have the same magnitude.

The study of the QLT asymptotics is possible without the full theory of specific heat because of the power law behaviour (1.6). Since the $\alpha$-variable is linear in temperature (1.2), an overall power of temperature factorizes out of the sum of individual summands (1.6), according to the hypothesis of independent and non-interfering velocities of sound (which is also one of the premises of the Debye theory),

$$
\begin{aligned}
C_m = \sum_i C_m^{(i)} &= \sum_i A_i k_B N_A \frac{\pi^4}{90} \frac{1}{\alpha_i^4} \\
&= k_B N_A \frac{\pi^4}{90} \sum_i A_i \left( \frac{k_B}{\hbar} \frac{a_i T}{v_i} \right)^4 \equiv \mathcal{A} k_B N_A T^4, \quad T < T_0,
\end{aligned}
\tag{1.7}
$$

where the sum is computed over all velocities of sound, $v_i$, enumerated by $i$. In this expression, we combined all physical properties of condensed matter (the lattice constant scaled along different crystallographic directions, the number of atoms per unit cell, the velocities of sound) and numerical coefficients into the single calibration constant $\mathcal{A}$ of (1.7).

The reference temperature $T_0$, which characterizes the threshold of the quasi-low temperature regime, is different for different materials, while the axiomatically derived universal function $\Theta(\alpha)$ is the same for all condensed matter systems. Therefore, according to (1.7), all condensed matter systems should behave as $T^4$ at sufficiently low temperatures. This sum over all independent velocities of sound is dominated by the *slowest* (minimal) velocity of sound. For the diamond cubic lattice, the lowest velocity is the transverse $v_5$, while for the face-centred cubic lattice it is $v_4$. For example, let us take the pressure (longitudinal) velocity $v_l$ and the shear (transverse) velocity $v_t$ for vitreous silica (table 2 below). According to (1.7), below the QLT threshold, the transverse wave contribution is 5.7 times greater than the $v_l$ contribution. Nevertheless, the longitudinal wave contribution is present at any temperature, even though its contribution is small in the QLT regime.

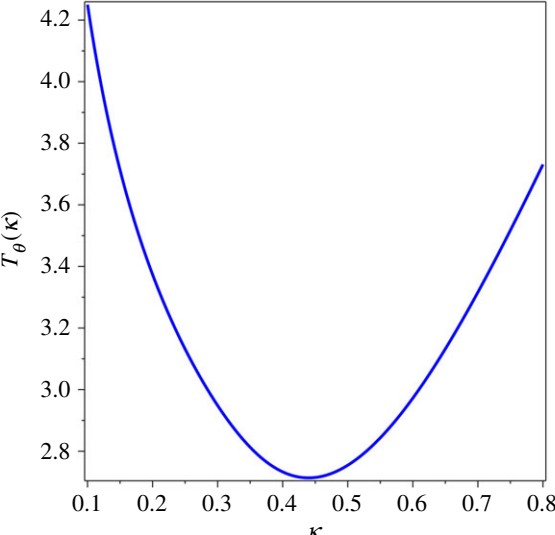

**Figure 4.** $T_\Theta(\kappa) = \kappa(8.31/\Theta(1/\kappa))^{1/3}$.

## 1.4. The threshold of the quasi-low temperature regime

We suggested [9] to use a combination *similar* to the Debye temperature [5] as a test function for analysing the power law behaviour of specific heat. The proposed function,

$$T_\Theta(T) \equiv T(R/C_m)^{1/3}, \tag{1.8}$$

differs from the Debye temperature, $T_D$, only by a numerical coefficient. The gas constant, $R$, is used to make the fraction dimensionless, so that we could have temperature on both axes. It is easy to see that $T_\Theta$ is perfectly suited for distinguishing two power laws under study, $T^3$ vs $T^4$. If the Debye theory were correct, then $T_\Theta = $ const, which should be a straight horizontal line of the graph of $T_\Theta$ vs $T$. However, if the specific heat behaviour is the fourth power of temperature, then $T_\Theta \propto T^{-1/3}$, within the QLT range. In a limited temperature range, the function $T^{-1/3}$ looks like a line with a negative slope. Indeed, for many decades [8], this graphical feature has been observed for the experimental Debye temperature, $T_D$, as discussed and illustrated by plots in §2. The minimum of (1.8) is the temperature where the power law of $T_\Theta$ changes. We proposed to use this temperature value, denoted by $T_0$, as the characteristic temperature of a specific material (at a given pressure, because the crystal structure and elastic properties of materials depend on pressure). The axiomatically derived thermal sum (1.4) can be used to make up a combination which exhibits a similar behaviour. The dimensionless function,

$$T_\Theta(\kappa) \equiv \kappa(8.31/\Theta(1/\kappa))^{1/3}, \tag{1.9}$$

of the inverse variable $\kappa = 1/\alpha$, which plays a role of dimensionless temperature, resembles the Debye temperature anomaly (figure 4).

The characteristic graph of the specific heat, $C_m/T^3$ vs $T$, is commonly used in solid-state physics. The Debye theory predicts $C_m/T^3 = $ const, which should be a straight horizontal line, this prediction is very different from the measurements of many experiments. The experimental function $C_m/T^3$ exhibits convexity in the QLT regime; in the literature on glassy matter, this 'hump' is associated with the so-called 'boson peak' [25]. The name came from the physics of Raman scattering, but this feature is apparently firmly associated with the specific heat behaviour discussed below [26].

The test function, $C_m/T^3$, serves the same purpose as $T_\theta$: it reveals the temperature at the inflection point, where the power law of specific heat changes. The power of $T$ of the left branch of this graph should be greater than three (we proposed it is four), while the power of its right branch must be less than three (we proposed it is the exponential damping). As shown with the experimental data below, the temperature of the minimum of $T_\theta(T)$ coincides with the temperature at the maximum of $C_m/T^3$; it is $T_0$.

Let us show that the function (1.4) leads to the same behaviour as $C_m/T^3$, when plotted by the inverse variable $\kappa = 1/\alpha$, which mimics $T$. By substituting $\alpha = 1/\kappa$ into $\Theta(\alpha)$ and plotting $\Theta(1/\kappa)/\kappa^3$ vs $\kappa$ we get figure 5. Let us emphasize that this kind of behaviour should be found for the specific heat of

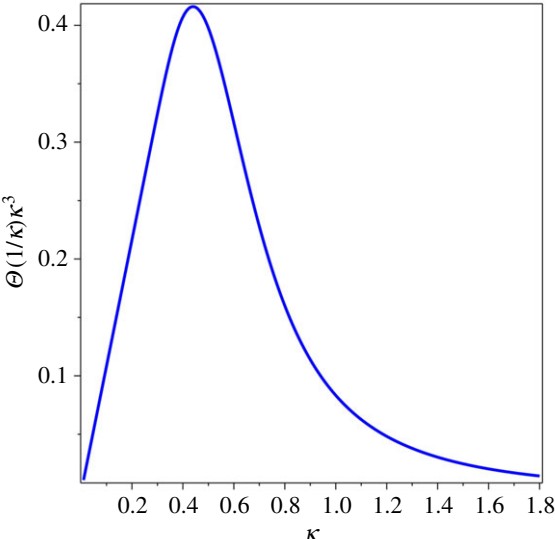

**Figure 5.** $\Theta(1/\kappa)/\kappa^3$.

any condensed matter systems, for any velocity of sound, if a system supports several different velocities of sound.

## 1.5. The temperature of the threshold of the QLT regime

The phenomenological characteristics discussed above follow from mathematical properties of the thermal sum (1.4). The location of the minimum of the graph of $T_\theta(\kappa)$ in figure 4 or the maximum of the graph of $\Theta(1/\kappa)/\kappa^3$ vs $\kappa$ in figure 5 can be found by numerical evaluation: $\alpha = 2.274$ or $\kappa = 1/\alpha = 0.440$. In a physical theory, this value is different for different classes of materials, and it was suggested [9] to use the dimensionless parameter,

$$\alpha_0 = \frac{\hbar v_{\min}}{k_B T_0 a}, \tag{1.10}$$

as a threshold characteristics of the QLT regime. Here we replaced the notation $\theta$ of [9] by $\alpha_0$ to avoid confusion with $\Theta(\alpha)$ and to show that (1.10) is a special value of $\alpha$. The characteristic temperature, $T_0$, in (1.10) is found by one of two testing functions described in the previous sections. It was conjectured that the constant $\alpha_0$ is the same for all materials with the same crystal lattice. Indeed, it was found [9] that (1.10) is almost the same for silicon, germanium, grey tin (whose sound velocity was *computed* from the neutron scattering data) and gallium arsenide, which all have the diamond (or zincblende) lattice.

Even though the contribution of the transverse sound wave dominates in the QLT regime, all sound waves exhibit the same behaviour shown in figures 4 and 5. The characteristic temperature $T_0$ is the result of combined contributions of all sound velocities. The leading contribution comes from $v_{\min}$, but other sound velocities spread and shift the minimum of $T_\theta$. Therefore, the parameter (1.10) is really *approximate*.

# 2. Experimental verification of the quasi-low temperature behaviour

We have studied [9] the QLT-specific heat of materials with the diamond lattice and concluded that those data confirm the quartic law of the field theory of specific heat. The materials were elements of the carbon group (diamond [27], silicon and germanium [28] and the zincblende lattice compound, gallium arsenide [29]). The data for several other materials from [29] were quantitatively studied and the results mentioned. To further demonstrate that this behaviour is not specific to the diamond lattice, we consider here the data for the face-centred lattice, silver chloride, AgCl and lithium iodide, LiI. Besides, the QLT behaviour is not limited to crystalline matter, it is also common for amorphous matter, vitreous silica is considered as an example.

**Table 1.** Physical properties of two fcc lattice compounds.

| material | $a$ (Å) | $\rho$ (g cm$^{-3}$) | $c_{11}$ (GPa) | $c_{12}$ (GPa) | $c_{44}$ (GPa) | $v_t$ (m/s) | $T_0$ (K) | $\alpha_0$ |
|----------|---------|---------------------|----------------|----------------|----------------|-------------|-----------|------------|
| AgCl | 5.546 | 5.699 | 75.85 | 39.08 | 6.892 | 1099 | 10.58 | 1.43 |
| LiI | 6.026 | 4.06 | 36.27 | 15.11 | 14.98 | 1823 | 14.26 | 1.71 |

## 2.1. Crystalline matter: face-centred cubic lattices of AgCl and LiI

In this section, we study the heat capacity of two compounds, silver chloride (AgCl) and lithium iodide (LiI). We remind that the Kopp–Newman rule, which states that the specific heat of a chemical compound is equal to the sum of specific heats of the compound's components, is not universal, i.e. it is not a law of physics. The Kopp–Newman rule does not hold for many materials, e.g. for solid binary antimonides [30]. In crystalline matter, an atom is a basic constituent of the system [31], therefore, we suggested [9] to assume that one mole of a compound is the Avogadro constant of its atoms, not molecules. This proposal means that we divide the specific heat values of the studied compounds by a factor of two.

Another note is on the difference between the molar specific heats at constant pressure vs constant volume, $C_p$ vs $C_v$. The heat capacity of condensed matter is measured at constant pressure, because the changing pressure changes the material's elastic properties and density, which define its thermal properties. Therefore, one should not correct the measured $C_p$ to $C_v$, even though this difference is small, as commonly done in the literature, e.g. [28]. Throughout the paper, we use the notation of the molar specific heat $C_m$ at constant pressure.

The slowest velocity of sound for the face-centred cubic lattice is the transverse wave propagating in the [100] crystallographic direction,

$$v_t = \left(\frac{c_{44}}{\rho}\right)^{1/2},$$  (2.1)

defined by the elastic constant $c_{44}$ and the density $\rho$. The published elastic constants for single crystals of silver chloride [32] give the computed velocity of sound in table 1. These experiments really measured the velocities by an ultrasound technique that were then converted to the elastic constants. Therefore, these are two equivalent descriptions of the crystal's elastic properties. It was measured that the elastic constant $c_{11}$ in the crystals of AgCl [33] grows almost twice with the temperature decreasing from 430°C to the room temperature. However, in the region of our interest, below 20 K, there are only two temperature points [32]. One can neglect the dependence on temperature of all three independent elastic constants, as it seems to be insignificant. The lattice constants (in ångström, $10^{-10}$ m) for AgCl and LiI in table 1 are taken from the reference book [31, p. 137], the density of AgCl (in g cm$^{-3}$) is from [32] and the density of LiI is from [34]. The elastic constants (in GPa) of LiI are given in the handbook [34] at room temperature ($c_{11} = 59.6$, $c_{12} = 36.2$, $c_{44} = 6.21$), so they were scaled to 20 K in the following way. From [33], the elastic constants for AgCl are also known at 293 K, and their temperature dependence is linear, which gives us a temperature scaling factor for each constant. We assume that the temperature dependence of the LiI constants is similar because both have the fcc lattice, thereby we computed the values $c_{ij}$ in table 1.

We use the datasets obtained by W.T. Berg (who used to work in the CRC Canada group of J.A. Morrison, which obtained other useful datasets [27,28]) for silver chloride and lithium iodide [35], in the temperature range from 2 to 20 K. This work is an excellent example of the quality of data (the stated experimental uncertainties are of the order of 1%) and the good description of experimental set-up. There also exists an older dataset for the temperature range from 15 to 292 K [36]. These data are confirmed by [37], although it did not publish newer data values.

Let us use the graph of $C_m/T^3$ vs $T$ for the analysis of the QLT regime in the AgCl specific heat. From this graph in figure 6 (in units $10^{-4}$ J mol$^{-1}$ K$^{-4}$), it is obvious that the QLT behaviour of specific heat does *not* obey the cubic power law at any temperature range. The plot in figure 6 is done by a solid line in order to visually compare it with the theoretical curve in figure 5. From this graph or directly from the $C_m/T^3$ values, we can find the characteristic temperature of AgCl, which is $T_0 = 10.6$ K. If we plot $T_\theta$ for AgCl specific heat data, figure 7 (in units $10^{-4}$ K), we find that the minimum of this graph coincides with the maximum of the graph in figure 6: it is the same $T_0$ as expected.

A

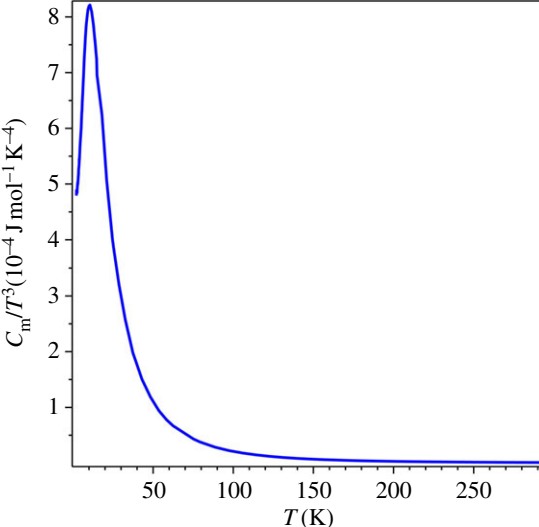

**Figure 6.** $C_m/T^3$ vs $T$ for AgCl [35,36].

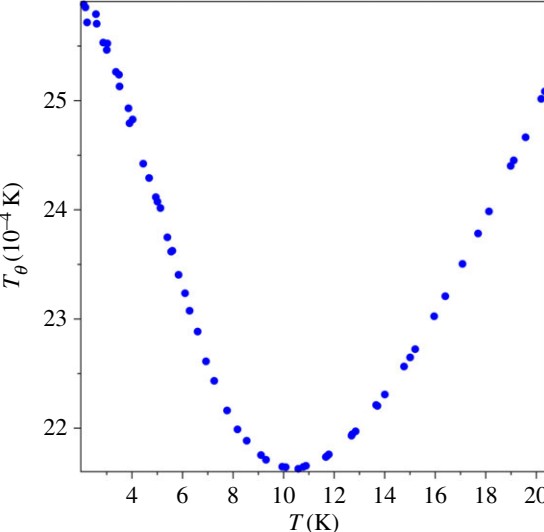

**Figure 7.** $T_\theta(T)$ for AgCl [35].

Now we test two statistical hypotheses for the specific heat data for $T < T_0$ (the left branch of the graph in figure 6): which power law these data obey, $T^3$ or $T^4$. We found that both hypotheses cannot be statistically rejected, but the quartic power one has a better $\chi^2$ statistic. We also fit the values of $C_m/T^3$ to the linear in temperature ansatz,

$$C_m/T^3 := d_0 + d_1 T. \tag{2.2}$$

The graphical result is displayed in figure 8 (in units $10^{-4}\,\mathrm{J\,mol^{-1}\,K^{-4}}$), and it confirms the proposed behaviour for the QLT regime (1.7). The coefficients of the fitting function (2.2) are $d_0 = 3.3 \times 10^{-4}\,\mathrm{J\,mol^{-1}\,K^{-4}}$ and $d_1 = 5.4 \times 10^{-5}\,\mathrm{J\,mol^{-1}\,K^{-5}}$. The fit (2.2) creates an impression that $C_m$ is finite at the absolute zero temperature since $d_0 > 0$. However, this theoretical formalism explicitly forbids $T \equiv 0$, equation (1.2). Nevertheless, the coefficient $d_0$ can be considered as an estimate for the surface specific heat, which is supposed to behave like $C_m^{(s)} \propto T^3$ according to the finite temperature field theory [9,13]. We leave the full study of two-dimensional systems for future work.

The data for the specific heat of lithium iodide are also from [35]. The statistical analysis is the same and its results are very similar to the ones for AgCl above, thus we do not display the corresponding graphs. This analysis gives the characteristic temperature for LiI $T_0 = 14.2\,\mathrm{K}$. The linear fit (2.2) of $C_m/T^3$ has the coefficients, $d_0 = 1.1 \times 10^{-4}\,\mathrm{J\,mol^{-1}\,K^{-4}}$ and $d_1 = 3.0 \times 10^{-5}\,\mathrm{J\,mol^{-1}\,K^{-5}}$.

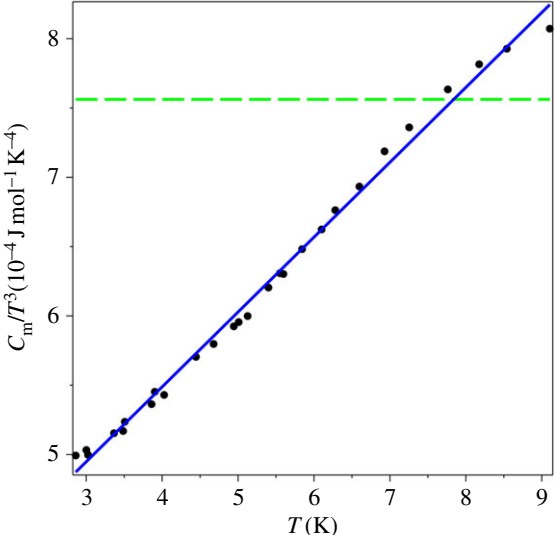

**Figure 8.** The QLT behaviour of $C_m/T^3$ vs $T$ for AgCl: dots—the experimental data [35], dashed line—the $T^3$ fit, solid line—the $T^4$ fit.

The parameters $\alpha_0$ computed for AgCl and LiI are somewhat different, 1.43 vs 1.71, which is probably explained by the extrapolation used for the elastic constants (the velocities of sound) at cryogenic temperatures. It is also possible that the type of crystal lattice does not define thermal properties alone. We also found that the graphs of $T_\theta(T)$ and $C_m/T^3$ vs $T$ for AgCl and LiI, after scaling with the corresponding temperatures $T_0$ and the maxima of these functions, align with each other only approximately, i.e. no 'master curve' is observed.

We have to criticize the data analysis of [35] which was done with the aim to confirm the Debye theory rather than to perform an unbiased statistical study. Specifically, the plotted figures and the fitting polynomials in [35] had pre-designed forms that were supposed to reveal the cubic law, while in reality they could be misleading. The graphs of $C_m/T^3$ vs $T^2$ for both materials were done only for temperatures below 8 K, thus they could not reveal the 'hump', e.g. figure 6, while $T^2$ on the horizontal axis served no useful purpose. Furthermore, the fitting polynomial was taken in the form, $C_m = aT^3 + bT^5 + cT^7$. Since too few points were taken for the LiI data ($T_0 = 14.26$ K), this polynomial gave an apparent visual agreement, but it clearly failed to match the AgCl data ($T_0 = 10.58$ K). In both cases, the uncertainty of the fitting coefficients was 25% for the coefficient $b$ and 50% for the coefficient $c$, which means the fitted parameters are statistically insignificant.

## 2.2. Amorphous matter: vitreous silica

By the derivation [9], the thermal function (1.4) should be applicable not only to crystalline but also to amorphous and liquid states of condensed matter. The anomalous (non-Debye) behaviour of specific heat of glasses has been known and studied for a long time [38,39]. Several models have been proposed to explain these anomalies thought to be specific to glasses, an active subject of theoretical and experimental studies [40]. Contrary to those models, the apparent anomalies are common with the specific heat behaviour of crystalline matter. This universal feature only demonstrates that the Debye theory of specific heat is wrong.

For example, the similarity in the shapes of the plots of $C_m/T^3$ for the amorphous and crystalline forms of the compound $Pd_{40}Cu_{40}P_{20}$ was noted in [41]. Namely, the measurements show that the temperature of the 'boson peak' is lower, while the value of $C_m/T^3$ at it is higher, in single crystals than in glass. This means that the crystalline form has a more 'glassy' behaviour than the glassy one. More recently, this observation was experimentally explored in the work [42] with single crystals of pentaphosphates of rare-Earth metals, gadolinium, $GdP_5O_{14}$ and neodymium, $NdP_5O_{14}$, and for glasses of the same compounds. It was found that the specific heat of glasses and crystals behaves qualitatively similarly, [42, fig. 3]. Below, we discuss this feature and propose its explanation with the field theory of specific heat. This theory for glasses should be simpler than for lattices, because it does have to deal with many contributions of different sound velocities in anisotropic matter. In this regard, the proposed theory satisfies the desired description by the pioneers of these studies, R.C.

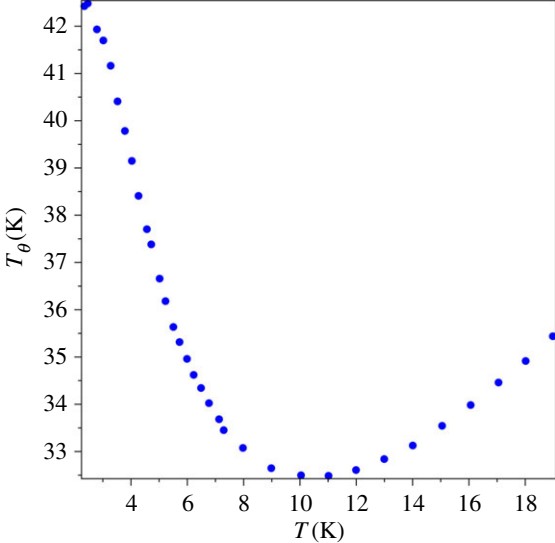

**Figure 9.** $T_\theta$ (K) for vitreous silica [44].

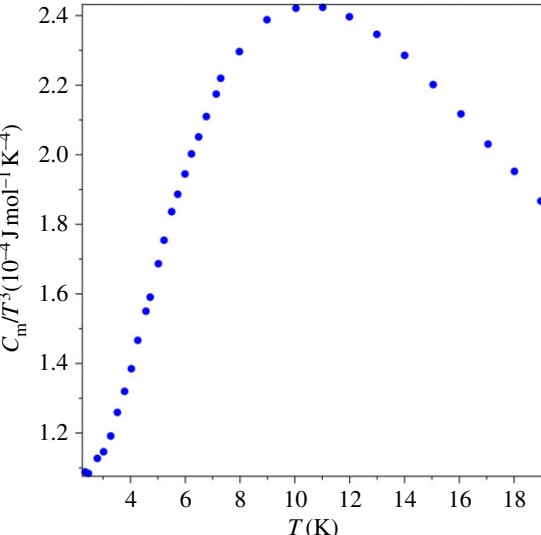

**Figure 10.** $C_m/T^3$ vs $T$ for vitreous silica [44].

Zeller and R.O. Pohl [43, p. 2039]: 'Any model aiming at an understanding of this anomaly has to be extremely simple in order to be equally applicable to a large number, if not to all, non-crystalline solids.'

The group of J.A. Morrison performed the measurements of the specific heat of vitreous silica and published the data [44]. The graph of $T_\Theta(T)$, figure 9, gives the characteristic temperature of vitreous silica, $T_0 = 10.5$ K. This value is cross-checked by the graph of $C_m/T^3$ vs $T$, figure 10 (in units $10^{-4}$ J mol$^{-1}$ K$^{-4}$). Both figures show that the specific heat of amorphous matter near the QLT regime displays the same physical behaviour as crystalline matter, cf. figures 6 and 7.

Applying the algorithm for extracting the QLT power law for specific heat above, we find that the specific heat of vitreous silica behaves like the fourth power of temperature, figure 11 (in units $10^{-4}$ J mol$^{-1}$ K$^{-4}$). The linear fit (2.2) of $C_m/T^3$ (the solid line in figure 11) has the coefficients $d_0 = 4.1 \times 10^{-5}$ J mol$^{-1}$ K$^{-4}$ and $d_1 = 2.5 \times 10^{-5}$ J mol$^{-1}$ K$^{-5}$.

The shear modulus of vitreous silica is 31.3 GPa [45], and the density is $\rho = 2.196$ g cm$^{-3}$, which give the transverse velocity of sound, $v_t = 3.73 \times 10^3$ m s$^{-1}$. The molar mass of this compound $M_{SiO_2} = 60.08$ g mol$^{-1}$ is divided by three, the number of atoms in its molecule, which corresponds to the assumption that intra- and inter-molecular distances are comparable in condensed matter. Then, the molar volume is found to be $V_m = 4.54 \times 10^{-29}$ m$^3$ mol$^{-1}$, which gives the average interatomic distance $a = 2.09 \times 10^{-10}$ m. Then, according to (1.8), we find the dimensionless parameter $\alpha_0 = 13.2$. This value is relatively close to this parameter for other glasses discussed below.

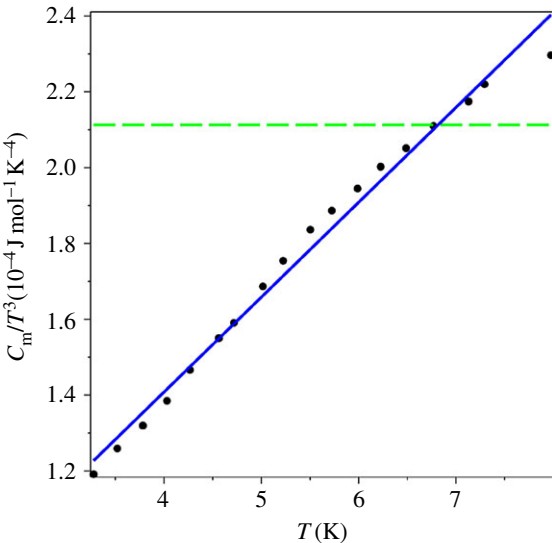

**Figure 11.** The QLT behaviour for vitreous silica: dots—the experimental data [44], dashed line—the $T^3$ fit, solid line—the $T^4$ fit.

**Table 2.** Physical properties of glasses.

| glass | $\rho$ (g cm$^{-3}$) | $M$ (g mol$^{-1}$) | $a$ (Å) | $v_t$ (m s$^{-1}$) | $T_0$ (K) | $\alpha_0$ |
|---|---|---|---|---|---|---|
| v. silica | 2.196 | 60.08 | 2.09 | 3775 | 10.04 | 13.2 |
| $B_2O_3$ | 1.838 | 69.62 | 2.32 | 1872 | 5.3 | 11.6 |
| $(Li_2O)_{0.14}(B_2O_3)_{0.86}$ | 2.071 | 64.06 | 2.22 | 2850 | 11.3 | 8.69 |
| $(Na_2O)_{0.16}(B_2O_3)_{0.84}$ | 2.122 | 68.40 | 2.25 | 2760 | 10.6 | 8.85 |
| $(K_2O)_{0.14}(B_2O_3)_{0.86}$ | 2.088 | 73.06 | 2.31 | 2301 | 8.3 | 9.17 |
| $(Cs_2O)_{0.14}(B_2O_3)_{0.86}$ | 2.484 | 99.33 | 2.41 | 1919 | 6.4 | 9.49 |

The work [26] contains general and elastic properties for several composite (molar proportions are indicated by subscripts) borate glasses, based on $B_2O_3$, that allow us to compute their constants $\alpha_0$ (data for the glass $(Na_2O)_{0.16}(B_2O_3)_{0.84}$ were taken from [46]). The densities, transverse velocities of sound and the characteristic temperatures are reproduced in table 2, together with the computed average interatomic distances $a$ and the thermal constants $\alpha_0$. The result appeared to be similar to the finding for the diamond lattices: $\alpha_0$ parameters for several materials in the same thermoelastic group are rather close, but not exactly equal. Their values of (1.10) slightly grow with the decreasing characteristic temperature $T_0$.

The experimental discovery of the fact that the graphs $C_m/T^3$ vs $T$ for glasses of the same kind but with different compositions, would form a 'master curve', i.e. collapse to a single curve, if the data were scaled by the temperatures and their values at the maxima of these graphs (at the 'boson peak' or $T_0$) was made by X. Liu & H.V. Löhneisen [47,48]. This fact displays another form of the scaling. The scaling phenomenon of the 'master curve' for such graphs was later analysed in the works [49,50]. This feature is commonly studied now [51].

In addition to the specific heat measurements, the physical nature of the 'boson peak' has been studied by other experimental techniques: inelastic neutron scattering, X-ray scattering and low-frequency Raman scattering. To understand its origin, the work [52] presented a comparison of the experimental results with vitreous glucose by far infrared (IR) spectroscopy in the terahertz region and by Raman scattering. The conclusion of [52] is that the 'boson peak' is observed at the frequency 1.4 THz at 14 K and that the IR spectroscopy dispersion curves coincide with the Raman scattering ones. Therefore, this work states that the 'boson peak' is the universal feature of glassy condensed matter consistently observed by different experimental techniques.

The excess of specific heat over the Debye cubic law expressed by $C_m/T^3$, at the quasi-low temperature is observed for many materials. In the field theory of specific heat [9], the quartic power

law is really universal for any form of condensed matter: crystalline, amorphous and liquid. Therefore, the 'boson peak' should be also common for any condensed matter. The fact that the 'boson peak' is not a special property of glasses was already suggested in several works. The work [53] studied the 'density of states' 'for glassy and crystalline polymorphs with matched densities'. It found that a glass and a corresponding crystal provide the same specific heat and suggest that any difference can be explained not by the difference in structures, but by the difference in densities. In our formalism, the density difference leads to the difference in sound velocities, which is one of the main parameters of a condensed matter system. Another research [51] studied the vitreous and permanently densified vitreous $GeO_2$, in comparison to its crystalline form, and concluded: 'our data give experimental evidence that glasses do not show any excess of vibrational modes when compared to their crystalline counterparts of similar mass density'. Incidentally, contrary to the graph in [51], which shows the $T^3$ behaviour (a flat line on the $C_m/T^3$ vs $T$ plot) of the specific heat data for tetragonal (rutile) crystal of $GeO_2$, and contrary to the reference cited therein, these data were obtained in the work [54], which found that for both glassy and crystalline matter 'excess of the limiting long-wave elastic $T^3$ component have been observed'.

The 'boson peak' can be observed not only below 10 K, as stated in [25]. The highest threshold of the QLT regime for solid matter is observed in diamond, $T_0 = 173$ K [9,27] due to diamond's unique elastic characteristics. The experimental and theoretical study of the 'boson peak' phenomenon in the specific heat for diamond (and other materials), via the Debye temperature function, was done by W. DeSorbo [55], even though he was not the first to note the similarity in the anomalous (non-Debye) behaviour of this function in general. The QLT threshold was reported at a lower temperature of 60 K, which could be attributed to the low-grade specimens ('fragmented boart') used for measurements. Physical theory gives no restriction on the absolute values of $T_0$, and pressure changes the elastic properties of condensed matter, therefore, the QLT regime may occur at even higher thermodynamic temperature.

## 3. Summary

— The physical idea of P. Debye about the connection between sound waves and specific heat of condensed matter is right, but its theoretical implementation is wrong.
— Elastic properties of condensed matter relevant to thermal phenomena are expressed by the group velocities of sound.
— The quasi-low temperature behaviour of specific heat for many condensed matter systems is the fourth power of temperature.
— The QLT regime of specific heat is defined not only by thermodynamic temperature, but also by the interatomic distances and by the sound velocities.
— The quartic power law is also supported by experimental data for the specific heat of the face-centred lattice compounds and vitreous silica.
— The non-Debye behaviour of $T_\theta(T)$ at the quasi-low temperature has the same physical nature as the 'boson peak' of $C_m/T^3$ vs $T$.
— The 'boson peak' is characterized by the temperature, below which the specific heat is dominated by the fourth power of temperature with the leading contribution from the transverse velocity of sound.

## 4. Conclusion

It is well known that traditional thermodynamics has limited applicability even in the physics of gases; thus, there are many theories of modified thermodynamics, e.g. [56]. Consequently it is also known that thermodynamics is not adequate for condensed matter systems. There are alternative theories describing thermal mechanical properties of condensed matter, e.g. [57]. The worst deficiency of thermodynamics of gases is that it does not include physical time, thus it is really thermostatics [24]. In the last decades, finite-time thermodynamics has been developed to cure this problem, e.g. [58]. The theoretical approach [13] used in this work is different from existing modifications of thermodynamics. It is based on geometrical formalism for the field theory and designed to be applied to condensed matter systems. Instead of modifying and amending thermodynamics for condensed matter systems, it re-derives it using principles and variables that are different from the ones used in traditional thermodynamics. The full theoretical structure and all experimental consequences of this theoretical proposal are yet to be discovered.

The main mathematical object of our calculations, the kernel of the evolution equation [9,13], is a direct descendant of the spectral sum computed by Peter Debye [6]. This subject of mathematical physics was started by Hermann Weyl in his work on the energy of thermal ('black body') radiation [59]. This mathematical problem evolved to become a large field of *spectral geometry*, with applications ranging from engineering to finance [12,60], let alone physical applications [11,21]. In fact, our result (1.3) is reminiscent of the Debye's spectral sum, because it is *inversely* proportional to the third power of dimensionless $\alpha$. The cut-off parameter $a$ is introduced in the defining proper time integral at its lower bound [9]. As can be seen from the variable's construction (1.2), $\alpha$ implies the maximum frequency of sound waves in condensed matter via the ratio of the sound velocity, $v$, and the lattice constant, $a$, which plays a role of the shortest wavelength. The spectral sum obtained by Debye is also proportional to the third power of the maximum frequency. Thus, the thermal functional $F(\alpha)$ computed via the kernel of the evolution equation can be viewed as some quasi-relativistic version of (but *not* an equivalent to) the spectral sum of sound waves eigen-frequencies.

Several physical mechanisms contribute to the total specific heat of condensed matter at low temperature, when electronic and magnetic phenomena are involved [61], which makes it difficult to distinguish them. Therefore, we focused on the simplest case of the heat capacity of condensed matter, its lattice dynamics expressed via sound waves. The basic statement of modern condensed matter physics is that thermal lattice energy of solid bodies is proportional to the third power of temperature at sufficiently low temperature. However, this is a theoretical prediction based on the belief that the thermodynamics of ideal (rarefied monoatomic) gases is universally applicable to condensed matter systems. The underlying statement for this work is that this belief is false and leads to wrong theories, whose predictions contradict experimental data.

The proposed thermodynamic (or rather thermostatic so far) formalism [13] embeds the concept of scale invariance in the condensed matter physics. As a result, the universal (scale-free) thermal function (1.4) was obtained. The high temperature behaviour of specific heat near the crystal–liquid phase transition temperature, named the Dulong–Petit (DP) limit, is quantitatively explained by the existence of the minimum characteristic length in all condensed matter systems, i.e. the lattice constant for crystalline matter and the average interatomic distance for amorphous and liquid matter. Even though this theory is not yet completed and calibrated, its low temperature behaviour could still be studied. We have analysed more experimental datasets that support the fourth power of temperature as the universal quasi-low temperature behaviour (1.7) of the acoustic energy contribution to the specific heat of condensed matter systems. The fourth power of $T$ in the QLT regime of specific heat corresponds to *four dimensions* of space–time [13] because in the finite temperature field theory, the leading power in (1.4) comes directly from the space–time dimension in the kernel of the evolution equation.

# 5. Discussion

## 5.1. The critique of the Debye theory

The heat capacity, like most other physical phenomena, can be described alternatively by models based on either *discrete* (oscillator) or *continuous* (field) variables. The history of the advent and rise of the quantum oscillator models of specific heat is described in the book [62]. Historically, the Einstein theory [63] was the first quantum model in condensed matter physics. It was found to be wrong already by P. Debye, but its supplementary use can still be found in the literature. More comments on the discrete models of specific heat are in the next subsection.

There are many discrepancies (often called anomalies) of experimental data with commonly used theories of specific heat. In the first paper [9], we revisited the idea of P. Debye about the specific heat as the energy density of standing sound waves in condensed matter. Even though we adopted the core physical idea, the Debye theory itself is theoretically inconsistent, as stated in [9, Appendix A] and fails to properly describe available experimental data, as is well known. Let us analyse the shortcomings of the Debye theory in more detail.

Let us first mention that the 1912 work of P. Debye was published in German [6] and it was not translated into English even for the book of his collected works [64] (its Russian translation is available in the collection [65]). Perhaps this is the reason why the description of the Debye theory in textbooks is often incomplete [5] and mixed up with other methods. In particular, the notions of the Brillouin zone [66] and the phonon [67] were introduced only in 1930 (although the name 'phonon'

was invented later [68, p. 24]). Even though it became common to discuss the heat capacity and other thermal phenomena in condensed matter in the language of phonons, as quasiparticles associated with sound waves, e.g. [69], we focus here on the original theory of P. Debye, which was formulated in coordinate space–time instead of phase space of quasiparticle theories, because our method works in space–time as well.

Part 2 of Debye's work [6] is entirely devoted to a mathematical problem, the calculation of the density of eigen-frequencies of standing sound waves in an elastic body. This calculation was done for a spherical body, the general problem for bodies of arbitrary shapes with smooth surface was soon solved by Weyl [70]. The history and the modern state of this problem and the corresponding large area of mathematical physics are reviewed in [71].

To build his theory of specific heat, Debye used several critical assumptions by postulating that (1) atoms within a solid body behave in a way similar to the atoms of a gas, which is quantitatively expressed by the equipartition theorem; (2) the energy of sound waves in an elastic body is proportional to the number of eigen-freqencies of the standing waves, which means one can calculate the body's acoustic spectrum to obtain the body's thermal energy.

The equipartition theorem was derived within thermodynamics of ideal gases and it states that each *mechanical* degree of freedom of a molecule contributes energy proportional to thermodynamic temperature. This is it: each atom is modelled as a mechanical system whose inertial centre can move along three spatial directions, and a polyatomic molecule of gas can possess in addition three rotational degrees of freedom. The equipartition theorem assigns the energy $k_B T$ to each mechanical degree of freedom of a molecule. It is well known that the predictions made using the equipartition theorem deviate significantly from experiment for real gases; thus, it is not exact, i.e. not a theorem, even within thermodynamics of gases.

The hypothesis about its applicability to atoms in condensed matter was at first justified by the Dulong–Petit law, but this law was found to be incorrect a hundred years ago. Nevertheless after introducing quantum theory into condensed matter physics, the equipartition theorem was transformed into another form. Each molecule is modelled now as a different mechanical system, the harmonic oscillator, whose discrete, due to the quantum hypothesis, modes of frequency $\omega$ contribute energy proportional to $\hbar\omega = k_B T$ to the total energy of the system. This is also a conjecture that cannot be directly verified by experiment; therefore, its validity rests on experimental verification of theoretical predictions made using this conjecture. The first and principal proof of this kind is the theory of specific heat. Experimental evidence is obvious beyond any doubt that these predictions are often in contradiction to measured quantities. Thus, the model of an atom as a harmonic (or anharmonic) oscillator cannot be viewed as confirmed.

This notwithstanding, the problem of finding specific heat of a crystal lattice composed of atoms was reduced to a different problem that belongs to elasticity theory, the theory dealing with mechanical properties of continuous matter, e.g. [72]. However, the crucial step of connecting thermodynamics of condensed matter, viewed as a lattice made of discrete constituents (atoms), with mechanics of condensed matter, viewed as an elastic medium, is a *postulate*, which looks unconvincing after unbiased consideration. Indeed, the theory of sound waves in elastic media does not know anything about lattice atoms (at least not in the form used by Debye; it does in our formalism through the minimum wavelength), because they belong to an entirely different, discrete, description of a system under study.

One more ill-defined procedure of Debye's derivation is the replacement of the discrete spectral sum by an integral over the continuous variable of frequency. Although mathematically this replacement is not correct, this procedure is commonly used in theoretical physics. A replacing continuous function can have mathematical properties which are principally different from those of its discrete counterpart, e.g. [73]. We suggested [9] that the thermal sum (1.4) always remains discrete and can only be evaluated analytically in the quasi-low temperature limit studied in this paper.

To bring into the *kinematic* solution, found within elasticity theory, the required physical dimensionality, energy (joule), Debye multiplied the whole expression, [6, eqn (74)], by the energy quantum, $\hbar\omega$. This step seemed to be quite natural and benign; however, it effectively increased the total power of the kinematic variable, the wave frequency, $\omega$, by one. Thus, by trying to extract the thermal energy quantity from the purely kinematic quantity, Debye changed the obtained mathematical solution. Upon the subsequent variable replacement, $\xi = \hbar\omega/(k_B T)$, [6, eqn (8)], this operation changes the total power of temperature as well. This is it: the frequency variable is first used to derive the *dimensionless* quantity, the total number of eigen-frequencies limited by some $\omega_{max}$. This number supposedly delivers (see the previous paragraph) the spectral density, which is

multiplied by the energy factor. This factor is not a part of the solved kinematic problem, but the frequency $\omega$ within it is used for the following integration anyway.

Debye compared his theoretical result with the measured specific heats for several elements [6, §I.4]; however, only one of them, diamond, was dielectric, while the remaining four were metals (aluminum, copper, silver and lead). The proof given in [6] was the matching of the Debye temperatures. No comparison with the cubic law was done, [6, p. 815]: 'After all, it would seem very desirable to examine the validity of the proportionality with $T^3$ in a series of observations of a single substance. Diamond would probably be the most suitable for such experiments.' We do not question the cubic law that could be observable in Debye's analysis for metals, because our statistical analysis of some datasets for metals also delivers the third power of temperature at low temperatures [9].

Interestingly enough, the controversy regarding the specific heat of diamond has a long history, since it was an element of choice for Einstein as well [63]. Continuing our analysis of the diamond's case started in [9], let us mention that natural diamonds have many impurities (despite their high value) and do not give experimenters much choice over the size and crystalline structure (because of their high value). As a result, there were no experimental data of sufficient precision for diamond until recently, as the best available data [27] showed convincingly [9]. Relatively recently, the high precision, comprehensive measurements were done at the Institute for Solid State Physics of the Max Planck Society [74]. These data show clearly the fourth power of temperature for diamond, as conjectured in [9]. In regard to artificial diamonds, they are polluted by the catalysis metallic material [74]. The analysis of the diamond's specific heat will be presented in a forthcoming paper.

## 5.2. The critique of the Born–von Karman theory

Beside the Debye theory, there are several other theories of specific heat. For instance, there is a phenomenological model of C.V. Raman based on the optical studies of crystals acoustic eigen-frequencies [75], but here we discuss the Born–von Karman theory of the vibrational spectra of lattices [5,76], which became a standard part of solid-state physics. However, it has several drawbacks, the fatal one is its failure to make any predictions, while its earlier versions could not even describe experimental data correctly.

The historical review and the details of the lattice vibration theory can be found in [76,77], while a brief look at the key developments is given by Max Born himself in [78]. The first work on the *mechanics* of atom vibrations was published at the same time as the Debye theory [79]. The second work [80] was a follow-up on the Debye theory [6]. At the present time, this is a much more complex and different construction, which is not really a theory, but a theoretical framework. This is it: the Born–von Karman theory does not give an explicit functional dependence among the system's physical observables under study, i.e. it only delivers the dispersion functions for acoustic waves in crystal, while the specific heat function is subsequently derived according to the Debye theory [76, §II.4].

Nevertheless, let us mention what M. Born and K. Huang wrote about the specific heat in the book on lattice dynamics [76]. The only comparison with experimental data is presented by [76, p. 41, fig. 4]. The book does not give any references for the data, and no quantitative study of the power law at low temperature is done. Furthermore, the figure consists of several curves, for various materials, whose high and low temperature pieces are detached. No curves for the whole range of temperatures are given, but it is well known that the Debye theory with its single variable does not work at all temperature regions. Therefore, while giving some indication of the validity of Debye's scaling hypothesis, this figure is not a confirmation of the quantitative behaviour of specific heat at low temperature. Like the purported graphical proofs of the $T^3$ law, given in Kittel's textbook [5], as discussed in [9], such a proof is also misleading.

The Born–von Karman theory relies on the widely accepted, but mathematically ambiguous, 'Born–von Karman boundary conditions' [5,76,81]. These conditions identify atoms on opposite faces of the crystal lattice as the same. This assumption makes up hypothetical cyclic chains of atoms and allows one to perform calculations. It is usually assumed that the number of atoms in the chain, $N$, is very large. This condition expressed as $N \to \infty$ is mathematically not correct, the assumed condition really means that a system is 'macroscopic', i.e. the number of its atoms is comparable with the Avogadro constant, $N/N_A = \mathrm{O}[1]$. The assumption is that for such long chains of atoms, the system's properties are independent of its (non-existent) boundaries. However, the expectation that atoms, which are $\mathrm{O}[N_A]$ lattice sites away from each other, behave completely independent of each other is not physical. From mechanical and thermal phenomena of solid-state systems, we know such systems transfer mechanical stress, thermal energy and electric current throughout the whole body.

The boundary of a condensed matter system is its essential physical component that serves as an external contact for studying its internal properties. As a matter of fact, boundaries can even define physical properties of solid-state systems discovered in recent decades, e.g. topological insulators. Since the Born–von Karman boundary conditions exclude the system's boundaries from physical consideration, they cannot be studied.

Furthermore, mathematically, the Born–von Karman boundary conditions replace a closed manifold with boundary by a compact manifold [82]. This replacement changes the topology of a physical system that could not be acceptable mathematically. The difference between physical solutions for such manifolds can be illustrated by solutions for the second-order differential operator (wave operator) on a membrane with free edge and a torus. Both manifolds are flat two-dimensional, but the latter one represents the Born–von Karman replacement for the former one, in two dimensions. Obviously, they support entirely different kinds of the operator's solutions as found in mathematical physics textbooks. The three-dimensional case is more difficult, because the three-dimensional torus, implicitly studied by Born, cannot be easily visualized.

In addition, the Born–von Karman theory is not applicable to disordered matter like glasses because it relies on the lattice order in its derivation. Therefore, the Born–von Karman theory cannot explain why the dispersion curves for ordered (crystalline) and disordered (amorphous) matter are quite similar. Indeed, it is well established now that the disorder of glassy and liquid matter does not damp the sound waves, as previously thought, and the dispersion curves for these forms of matter are similar to the ones for crystalline matter [83]. This fact was discovered with the help of powerful synchrotron radiation sources in the experiments done on liquid metals. The work [83] suggests that acceptance of this theoretical idea of J. Frenkel [84] (which still had to be experimentally verified) was much delayed by the prevailing concepts of harmonic oscillators and ideal lattices, which were used to build the lattice dynamics theory. As stated in [83], 'most important changes of thermodynamic properties of the disordered system are governed only by its fundamental length, the interatomic separation'. This statement agrees with our proposal, which includes the (average) interatomic distance as one of its fundamental parameters that enters the theory's variable (1.2).

Predictions of the Born–von Karman theory have been criticized by experimental physicists for decades. In particular, J.E. Desnoyers and J.A. Morrison wrote in their work on the diamond's specific heat [27] about theoretical results of H.M.J. Smith [85], who elaborated the Born–von Karman theory to obtain the diamond's frequency spectrum: 'it is evident that they depart seriously from $\Theta_D$ as calculated from the measured heat capacities. ... reservations may be held about the assumption of central forces for second neighbour interactions. It may prove useful to try in preference a five parameter theory assuming quite general forces between both first and second neighbours.' In other words, the simple model of the nearest neighbour interactions fails, however, and it can be made acceptable by introducing more free (fitting) parameters for the further neighbour interactions. The values of the calibration parameters are found by fitting the experimental data. Such extensions of the original theory could make it agree with the data, but this agreement is really a *postdiction*, while many calibration parameters render a theory useless for predictions.

The pioneering experiments of B.N. Brockhouse & P.K. Iyengar on thermal neutron scattering in single crystals of germanium (and later of other materials) made them conclude [86]: 'the results of these neutron experiments thus seem to be in good agreement with other experimental data in the literature, but are not in agreement with any very simple model of the interatomic forces. This lack of a suitable model is disturbing. The Born–von Karman calculation might be extended to more and more neighbours until agreement is reached, but then the satisfaction of Born's identity must be dismissed as an accident.' The Born identity mentioned is a special constraint on the elastic constants, which was used by Born to prove the condition that only nearest neighbour interactions should be taken into account [78]. Experiments showed decisively that both assumptions are false, the Born identity does not hold and long-range interatomic interactions should be accounted for.

Brockhouse said in his Nobel Prize lecture [87, p. 742]: 'the Born–von Karman theory itself can be taken to be phenomenological with innumerable possibilities for parameters, and thus hardly to be capable of refutation. If taken literally, as involving forces between ions in the crystal, then already these early results show surprisingly long-range behaviour for the interatomic force system.' A theory that cannot be falsified, because it can indefinitely be adjusted to fit any data, is not a theory. In recent years the scientific polemic about this issue was rather hot in regards to other *irrefutable* theories, notably in cosmology and high energy theory [88], even though this problem is not limited to those fields.

On the other hand, taking this 'surprisingly long-range behaviour' of many-body systems (crystals) discovered by the nuclear physics method to its logical limit, should all interactions among atoms be

taken into account? Indeed, all atoms interact with each other indirectly, through the medium they create. Then, physics of a system under study becomes *non-local*, which is in agreement with the observed failure of models based on local interactions. Further, it is natural to take as the physical characteristics of a non-local model of condensed matter the group velocity of sound. Indeed, the propagating or standing wave in a body represents a phenomenon of the *collective* behaviour of matter's constituents, and its properties reflect the features of interatomic interactions important for observable physical effects.

The work [89] deals with both the neutron scattering measurements of the crystal frequencies and the atomistic modelling of crystal physical properties. Its best-fitting model has as many as 13 adjustable parameters and includes the interatomic interaction with the second nearest neighbours. Too many calibration parameters lead to the situation when 'the 13-parameter model, while providing an improved fit to the data, gives values for several of the parameters that have no obvious physical significance' [89]. The number of the interatomic force constants kept increasing along with the increase of computer power and the precision of experimental data. In work [90], the phonon spectra of diamond, Si, Ge and $\alpha$-Sn (the elements also studied in our work [9]) were obtained using real-space interatomic force constants up to the 25th nearest neighbours. The coincidence with the experimental data was found to be excellent; however, one may wonder if the 25th nearest neighbour is too far for a local interaction, and what is the *practical* value of the theory that needs that many free parameters? In theoretical physics, fitting sufficiently many calibration parameters by experimental data can make any theory agree with available data, but such a theory may not be capable of forecasting new (unknown yet) physical phenomena.

## 5.3. The bicentennial of the work of A.T. Dulong & P.L. Petit

By coincidence, the year 2017 marked two centuries since the publication of the pioneering works of P.L Petit & A.T. Dulong [91,92], who thereby initiated the modern physics of thermal phenomena. It is remarkable that the work of J. Stefan on the thermal radiation law [93] was based on the old experimental data obtained in their work [92] on cooling of the heated bodies. The first part of the same investigation of Petit & Dulong [91] preceded their famous work on the Dulong–Petit law [4] mentioned above. Thus, the heat capacity of solid bodies and their thermal radiation were intrinsically (experimentally and theoretically) connected from the advent of both fields of physical study. This case well represents the eternal evolution of scientific research: both theories of Dulong & Petit (the 1817 law of thermal radiation and the 1819 law of specific heat) were used for long periods of time, but eventually proved to be wrong (the former one in 1879 [93], the latter one in 1907 [63]). However, these theories and especially the corresponding experimental data, albeit imprecise, paved the way for new theories, confirmed by better experiments. We believe that the time to move further on is long overdue.

## 5.4. The threshold of the QLT regime

No scientific idea is ever entirely new. We found that the dimensionless characteristic parameter for specific heat, similar to equation (1.10), was proposed by M. Blackman in his work on the anomalous vibrational spectra [94]. He shaped the ideas about systematic corrections to the 'anomalous' (non-Debye) Debye temperatures as [94, eqn (20)], which can be identically transformed to the equation,

$$T_D = \Theta' \frac{\hbar}{k_B} \left(\frac{c_{12}}{\rho}\right)^{1/2} \left(\frac{N_A}{V_m}\right)^{1/3}, \tag{5.1}$$

where $V_m$ is the molar volume. The expression (5.1) contains the velocity of sound, $v_{12}$, and an estimate for the interatomic distance via the average volume per lattice atom. The Blackman's relation introduced the characteristic parameter $1/\Theta'$ similar to our $\alpha_0$, while the Debye temperature $T_D$ features in (5.1) because $T_0$ of (1.10) denotes the temperature, where the slope of $T_\theta$ is zero, i.e. $C_m$ obeys the Debye cubic law at this point. The Blackman's parameter $1/\Theta'$ characterizes certain classes of materials analysed in [94].

A physical expression similar to (1.10) and (5.1) was also derived and verified in [41]. For the fcc and bcc crystals it suggested the form $\omega^* = 4(c_{44}/\rho)^{1/2}/a$, i.e. this is the ratio of the shear velocity and the lattice constant. Since the frequency $\omega^*$ of the 'boson peak' is proportional to the characteristic temperature $T_0$, this combination is really identical to our parameter $\alpha_0$, up to a numerical coefficient. The fact that the 'boson peak' temperature, $T_0$, is almost linearly correlated with the shear acoustic velocity for many glasses was empirically discovered in [95] and further confirmed in [26,96].

In [9], we discussed the specific heat data for solid argon and krypton [97], which clearly show the $T^4$ behaviour below correspondingly $T_0 = 8.0$ K and $T_0 = 6.0$ K. Later, the group of N.E. Phillips measured also thermodynamic properties of solid helium $^4$He [98], which crystallizes to the body-centred cubic lattice below 2 K. In that paper, graphical and quantitative analysis was performed on experimental data tabulated for several measurement runs, with different molar quantities of the formed solid phase. The authors, who aimed at measuring the constant volume specific heat, $C_v$, clearly found from [98, fig. 20] ($C_v/T^4$ vs $T$) that the specific heat of the bcc solid $^4$He 'apart from the pretransition anomaly, is approximately proportional to $T^4$', below the temperature about 1.6 K. This temperature of the transition to the bcc phase depends on the pressure. Let us note that the type of graph, $C_m/T^4$ vs $T$, could be more suitable to demonstrate the validity of the fourth power law, but the traditional graph $C_m/T^3$ vs $T$ is more convenient for determining the value of $T_0$. We analysed some data from [98] with the algorithms presented above and confirm the above conclusion of N.E. Phillips *et al.* about the specific heat law of the bcc solid helium. Therefore, the transition temperature of about 1.6 K is currently the lowest found characteristic temperature of the QLT threshold, i.e. the 'boson peak' feature is apparent even in such extraordinary condensed matter systems.

## 5.5. The cubic law in the specific heat experimental data

First of all, the main theoretical conclusions about the behaviour of specific heat were drawn before the high precision measurement data became available by the mid-twentieth century. As a result, no statistical analyses were done, as comparison to theories was performed either by graphical methods or by the simple fitting with the goal to re-confirm the Debye law. We advocate applying an unbiased analysis based on the statistical significance to all available specific heat data, as commonly done in other areas of physics, where experimental data are much more abundant, like observational cosmology or accelerator physics. Therefore, the main statement of this paper is that experimental data for the specific heat of some materials, at some temperature ranges, fit the fourth power of temperature, and this fit is statistically more preferable than the cubic power law.

Indeed, statistical significance, as a chief mathematical tool for distinguishing competing scientific hypotheses, is the only way to select the most acceptable theory. Therefore, when the $T^3$ power or a more general, odd powers, polynomial is used as a test function, this function, and *any* other function, can always be fitted. However, the question of how physically meaningful this fit is can only be answered with help of standard deviations for the coefficients of the fitting polynomial and the value of chi-square, $\chi^2$, statistic (and/or other statistics) of statistical significance of that fit, e.g. [10, §39]. Regretfully, the validity of the Debye cubic power law of specific heat was never questioned, and it was never scrutinized by statistical means.

Nevertheless, is the cubic power of temperature anywhere in experimental data? Indeed, it is; the characteristic temperature (§1.5) is a threshold between the specific heat regimes of the fourth power law (the quasi-low temperature range) and the damped exponential law (the intermediate temperature range) of the general function (1.4). The maximum of $C_m/T^3$ vs $T$ corresponds to the point where the $T^3$ law always holds exactly, i.e. the slope of the graph at this point is zero, as it should be for the Debye law.

As mentioned in [9], the cubic power law at low temperatures can be seen in the specific heat data for metals. To further expand that list, we tested the data for yttrium [99], whose specific heat definitely exhibits the $T^3$ behaviour *above* 10 K. We cannot apply the present theoretical formalism to electronic phenomena yet. Therefore, let us just mention this empirical finding of the cubic power law, in contrast to the fourth power law found above, without making any conjecture about its physical origin.

## 5.6. The scaling in the specific heat phenomena

The scaling in the specific heat phenomena was first explicitly formulated by Peter Debye in his pioneering work [6, p. 793]: 'If one calculates the temperature $T$ as a multiple of the substance's characteristic temperature $\Theta$, then the specific heat for all (monatomic) bodies is represented by the same curve, in other words, the specific heat of a monatomic body is a universal function of the ratio $T/\Theta$.' This scaling hypothesis was realized in the Debye specific heat function, the dimensionless function (multiplied by the dimensionful factor $3R$) of the dimensionless variable $T/\Theta$. Debye's expectation that the specific heat of all solid (monatomic) materials would possess the same universal function was too general. Our proposal [9] is that all materials with the same crystal lattice type should match the same specific heat function. The built-in assumption that the universal Debye

function depends on a single variable was quickly destroyed by experiment. Even though the Debye theory, in our view, does not pass the crucial test of the quasi-low temperature behaviour, as illustrated above by experimental data, he should rightly be credited for the foresight of the scaling phenomenon in specific heat.

It is obvious that the intrinsic scaling property of the condensed matter's specific heat should be evident regardless of an experimental method. The isomorphism of the specific heat functions of materials belonging to the same class, e.g. the diamond type crystal lattice, is just one way to see this scaling. However, physical properties of condensed matter that define its specific heat exhibit this scaling in different forms. The hypothesis that the lattice dynamics of all homopolar crystals with the diamond lattice are homologous was expressed by Kucher [100] who tested it with silicon and diamond data. Her work used the the dimensionless frequency variables introduced by Tolpygo [101]. Since then, the study of this scaling was greatly advanced. It was discovered by Nilsson and Nelin [102], who conducted a series of experiments on the thermal neutron scattering in crystals, that the phonon dispersion curves of germanium and silicon nearly coincide. This idea was further extended to include also the zincblende lattice compounds by Reyes & Poniatowski [103]. This finding agrees with our proposal that all elements and compounds with the diamond type lattice belong to the same scaling class. This scaling is a reflection of the similarity of elastic properties of diamond and zincblende lattice materials as mentioned in [9, §III-D].

Let us conclude by highlighting the pioneering work of J.D. van der Waals [104], who apparently was the first scientist to discover the general and profound property of physical theories, the scaling. He introduced dimensionless thermodynamic variables by scaling the volume, pressure and temperature of gases with their corresponding values at the critical point. This operation let him obtain the dimensionless equation of state, which is applicable *universally* to various real gases. This discovery has led over the past century to the fruitful development of the scaling theory of critical phenomena. The scale-free equation of state derived by van der Waals was the first of this kind of equation of physics. The present proposal of the field theory of specific heat of condensed matter is the next step in the development of the scaling principle.

Data accessibility. This paper has no data or associated material.

Competing interests. I have no competing interests.

Funding. There are no funders to report for this work.

Acknowledgements. I am grateful to the Max Planck Institute for Gravitational Physics (Albert Einstein Institute) at Potsdam-Golm, Germany for the support and hospitality during visits.

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
