## [Reviewer comments · Royal Society Open Science]

Review History

RSOS-171285.R0 (Original submission)

Review form: Reviewer 1

Is the manuscript scientifically sound in its present form?

No

Are the interpretations and conclusions justified by the results?

No

Is the language acceptable?

Yes

Is it clear how to access all supporting data?

Yes

Do you have any ethical concerns with this paper?

No

Have you any concerns about statistical analyses in this paper?

Yes

Recommendation?

Major revision is needed (please make suggestions in comments)

Comments to the Author(s)

Please see attached file (Appendix A).

Decision letter (RSOS-171285.R0)

11-Dec-2017

Dear Dr Gusev,

The editors assigned to your paper ("The quasi-low temperature behaviour of specific heat") have now received comments from reviewers. We would like you to revise your paper in accordance with the referee and Associate Editor suggestions which can be found below (not including confidential reports to the Editor). Please note this decision does not guarantee eventual acceptance.

Please submit a copy of your revised paper within three weeks (i.e. by the 03-Jan-2018). If we do not hear from you within this time then it will be assumed that the paper has been withdrawn. In exceptional circumstances, extensions may be possible if agreed with the Editorial Office in advance. We do not allow multiple rounds of revision so we urge you to make every effort to fully address all of the comments at this stage. If deemed necessary by the Editors, your manuscript will be sent back to one or more of the original reviewers for assessment. If the original reviewers are not available, we may invite new reviewers.

- Data accessibility

If you wish to submit your supporting data or code to Dryad (<http://datadryad.org/>), or modify your current submission to dryad, please use the following link:
<http://datadryad.org/submit?journalID=RSOS&manu=RSOS-171285>

- Competing interests

- Authors' contributions

- Acknowledgements

- Funding statement

Please note that Royal Society Open Science will introduce article processing charges for all new submissions received from 1 January 2018. Charges will also apply to papers transferred to Royal Society Open Science from other Royal Society Publishing journals, as well as papers submitted as part of our collaboration with the Royal Society of Chemistry (<http://rsos.royalsocietypublishing.org/chemistry>). If your manuscript is submitted and accepted for publication after 1 Jan 2018, you will be asked to pay the article processing charge, unless you request a waiver and this is approved by Royal Society Publishing. You can find out more about the charges at <http://rsos.royalsocietypublishing.org/page/charges>. Should you have any queries, please contact openscience@royalsociety.org.

on behalf of Dr Robert Young (Associate Editor) and Miles Padgett (Subject Editor)
openscience@royalsociety.org

Associate Editor's comments (Dr Robert Young):

The reviewer has raised a number of issues that need to be addressed by major revisions of this manuscript. In its current form we cannot support the conclusions that are made.

Comments to Author:

Reviewers' Comments to Author:
Reviewer: 1

Comments to the Author(s)
Please see attached file.

Author's Response to Decision Letter for (RSOS-171285.R0)

See Appendix B.

RSOS-171285.R1 (Revision)

Review form: Reviewer 1

Is the manuscript scientifically sound in its present form?

No

Are the interpretations and conclusions justified by the results?

No

Is the language acceptable?

Yes

Is it clear how to access all supporting data?

Not Applicable

Do you have any ethical concerns with this paper?

No

Have you any concerns about statistical analyses in this paper?

I do not feel qualified to assess the statistics

Recommendation?

Major revision is needed (please make suggestions in comments)

Comments to the Author(s)

The author has added an extended discussion section that is very helpful and improves the paper considerably. In particular, it includes the acknowledgement that the T^3 law for the heat capacity of solids has indeed been well established in the case of metals. It also includes a scholarly critique of the various assumptions that are involved in the standard analyses of the thermal properties of materials, including the transition from the discrete to the continuous regime, the use of boundary conditions that may misrepresent the topology of a material, and more.

Despite these improvements the crucial question of how these various effects are relevant to the theory proposed by the author (that the T^3 law for the heat capacity must be augmented with an extra factor of temperature, to give rise to a T^4 law) has not been clarified. The present reviewer cannot confirm the theoretical basis for the T^4 law and is uncertain whether the experimental evidence presented has ruled out the applicability of the conventional T^3 law in the low temperature limit even in the case of insulators.

I believe, however, that the author has raised interesting theoretical points and highlighted a number of observations that may question our understanding of the thermal properties of certain insulators at low temperatures. One possibility would be for the author to present his critique alone leaving to a future paper a discussion of how the apparent anomalies in the thermodynamic data can be accounted for in terms of a new theoretical description. I believe that this future paper would have to make the case for the T^4 law for the heat capacity that can be persuasive in terms of both analytical and physical arguments. In particular, the future paper would need to explain both analytically and physically precisely which shortcoming of current theory leads to an allegedly misleading T^3 law and how the correction of this shortcoming yields an additional factor of T in the heat capacity. I encourage the author to develop further his strikingly original ideas.

Review form: Reviewer 2**Is the manuscript scientifically sound in its present form?**

Yes

Are the interpretations and conclusions justified by the results?

Yes

Is the language acceptable?

Yes

Is it clear how to access all supporting data?

Not Applicable

Do you have any ethical concerns with this paper?

No

Have you any concerns about statistical analyses in this paper?

No

Recommendation?

Major revision is needed (please make suggestions in comments)

Comments to the Author(s)

Please, see the file attached (Appendix C).

Decision letter (RSOS-171285.R1)

21-May-2018

Dear Dr Gusev:

Manuscript ID RSOS-171285.R1 entitled "The quasi-low temperature behaviour of specific heat" which you submitted to Royal Society Open Science, has been reviewed. The comments of the reviewer(s) are included at the bottom of this letter.

Please submit a copy of your revised paper within three weeks (i.e. by the 13-Jun-2018). If we do not hear from you within this time then it will be assumed that the paper has been withdrawn. In exceptional circumstances, extensions may be possible if agreed with the Editorial Office in advance. We do not allow multiple rounds of revision so we urge you to make every effort to fully address all of the comments at this stage. If deemed necessary by the Editors, your manuscript will be sent back to one or more of the original reviewers for assessment. If the original reviewers are not available we may invite new reviewers.

- Ethics statement

If your study uses humans or animals please include details of the ethical approval received, including the name of the committee that granted approval. For human studies please also detail

whether informed consent was obtained. For field studies on animals please include details of all permissions, licences and/or approvals granted to carry out the fieldwork.

- Data accessibility

- Competing interests

- Authors' contributions

- Acknowledgements

- Funding statement

Please note that Royal Society Open Science will introduce article processing charges for all new submissions received from 1 January 2018. Charges will also apply to papers transferred to Royal Society Open Science from other Royal Society Publishing journals, as well as papers submitted as part of our collaboration with the Royal Society of Chemistry (<http://rsos.royalsocietypublishing.org/chemistry>). If your manuscript is submitted and accepted for publication after 1 Jan 2018, you will be asked to pay the article processing charge, unless you request a waiver and this is approved by Royal Society Publishing. You can find out more about the charges at <http://rsos.royalsocietypublishing.org/page/charges>. Should you have any queries, please contact openscience@royalsociety.org.

Once again, thank you for submitting your manuscript to Royal Society Open Science and I look

forward to receiving your revision. If you have any questions at all, please do not hesitate to get in touch.

Kind regards,
Andrew Dunn
Royal Society Open Science
openscience@royalsociety.org

on behalf of Dr Robert Young (Associate Editor) and Miles Padgett (Subject Editor)
openscience@royalsociety.org

Associate Editor Comments to Author (Dr Robert Young):

Associate Editor: 1

Comments to the Author:

Please read the comments made the reviewer carefully, unfortunately we cannot publish your manuscript in its current form.

Associate Editor: 2

Comments to the Author:

(There are no comments.)

Reviewer comments to Author:

Reviewer: 1

Comments to the Author(s)

The author has added an extended discussion section that is very helpful and improves the paper considerably. In particular, it includes the acknowledgement that the T^3 law for the heat capacity of solids has indeed been well established in the case of metals. It also includes a scholarly critique of the various assumptions that are involved in the standard analyses of the thermal properties of materials, including the transition from the discrete to the continuous regime, the use of boundary conditions that may misrepresent the topology of a material, and more.

Despite these improvements the crucial question of how these various effects are relevant to the theory proposed by the author (that the T^3 law for the heat capacity must be augmented with an extra factor of temperature, to give rise to a T^4 law) has not been clarified. The present reviewer cannot confirm the theoretical basis for the T^4 law and is uncertain whether the experimental evidence presented has ruled out the applicability of the conventional T^3 law in the low temperature limit even in the case of insulators.

I believe, however, that the author has raised interesting theoretical points and highlighted a number of observations that may question our understanding of the thermal properties of certain insulators at low temperatures. One possibility would be for the author to present his critique alone leaving to a future paper a discussion of how the apparent anomalies in the thermodynamic data can be accounted for in terms of a new theoretical description. I believe that this future paper would have to make the case for the T^4 law for the heat capacity that can be persuasive in terms of both analytical and physical arguments. In particular, the future paper would need to explain both analytically and physically precisely which shortcoming of current theory leads to an allegedly misleading T^3 law and how the correction of this shortcoming yields an additional factor of T in the heat capacity. I encourage the author to develop further his strikingly original ideas.

Reviewer: 2

Comments to the Author(s)
Please, see the file attached

Author's Response to Decision Letter for (RSOS-171285.R1)

See Appendix D.

RSOS-171285.R2 (Revision)

Review form: Reviewer 1

Is the manuscript scientifically sound in its present form?

No

Are the interpretations and conclusions justified by the results?

No

Is the language acceptable?

Yes

Is it clear how to access all supporting data?

Not Applicable

Do you have any ethical concerns with this paper?

No

Have you any concerns about statistical analyses in this paper?

I do not feel qualified to assess the statistics

Recommendation?

Accept as is

Comments to the Author(s)

Although the author does not wish to take up my principal recommendation in the last report, he has made considerable effort to incorporate the suggestions made by the second reviewer. If the second reviewer is satisfied with the modified manuscript, I would not wish to stand in the way of publication.

Review form: Reviewer 2

Is the manuscript scientifically sound in its present form?

Yes

Are the interpretations and conclusions justified by the results?

No

Is the language acceptable?

Yes

Is it clear how to access all supporting data?

Not Applicable

Do you have any ethical concerns with this paper?

No

Have you any concerns about statistical analyses in this paper?

No

Recommendation?

Major revision is needed (please make suggestions in comments)

Comments to the Author(s)

Please, see the file attached (Appendix E).

Decision letter (RSOS-171285.R2)

31-Aug-2018

Dear Dr Gusev:

Manuscript ID RSOS-171285.R2 entitled "The quasi-low temperature behaviour of specific heat" which you submitted to Royal Society Open Science, has been reviewed. The comments of the reviewer(s) are included at the bottom of this letter.

Please submit a copy of your revised paper before 23-Sep-2018. Please note that the revision deadline will expire at 00.00am on this date. If we do not hear from you within this time then it will be assumed that the paper has been withdrawn. In exceptional circumstances, extensions may be possible if agreed with the Editorial Office in advance. We do not allow multiple rounds of revision so we urge you to make every effort to fully address all of the comments at this stage. If deemed necessary by the Editors, your manuscript will be sent back to one or more of the original reviewers for assessment. If the original reviewers are not available we may invite new reviewers.

To revise your manuscript, log into <http://mc.manuscriptcentral.com/rsos> and enter your Author Centre, where you will find your manuscript title listed under "Manuscripts with Decisions." Under "Actions," click on "Create a Revision." Your manuscript number has been

appended to denote a revision. Revise your manuscript and upload a new version through your Author Centre.

- Ethics statement

- Data accessibility

- Competing interests

- Authors' contributions

- Acknowledgements

- Funding statement

Please note that Royal Society Open Science charge article processing charges for all new submissions that are accepted for publication. Charges will also apply to papers transferred to Royal Society Open Science from other Royal Society Publishing journals, as well as papers submitted as part of our collaboration with the Royal Society of Chemistry (<http://rsos.royalsocietypublishing.org/chemistry>). If your manuscript is newly submitted and subsequently accepted for publication, you will be asked to pay the article processing charge, unless you request a waiver and this is approved by Royal Society Publishing. You can find out more about the charges at <http://rsos.royalsocietypublishing.org/page/charges>. Should you have any queries, please contact openscience@royalsociety.org.

on behalf of Dr Robert Young (Associate Editor) and Prof. Miles Padgett (Subject Editor)
openscience@royalsociety.org

Associate Editor Comments to Author (Dr Robert Young):

Associate Editor: 1

Comments to the Author:

Following review, there are major points raised by the reviewers that remain outstanding - please address these.

Reviewer comments to Author:

Reviewer: 2

Comments to the Author(s)

Please, see the file attached

Reviewer: 1

Comments to the Author(s)

Although the author does not wish to take up my principal recommendation in the last report, he has made considerable effort to incorporate the suggestions made by the second reviewer. If the second reviewer is satisfied with the modified manuscript, I would not wish to stand in the way of publication.

Author's Response to Decision Letter for (RSOS-171285.R2)

See Appendix F.

RSOS-171285.R3 (Revision)

Review form: Reviewer 2

Is the manuscript scientifically sound in its present form?

Yes

Are the interpretations and conclusions justified by the results?

Yes

Is the language acceptable?

Yes

Is it clear how to access all supporting data?

Not Applicable

Do you have any ethical concerns with this paper?

No

Have you any concerns about statistical analyses in this paper?

No

Recommendation?

Accept as is

Comments to the Author(s)

See attached file (Appendix G).

Decision letter (RSOS-171285.R3)

04-Jan-2019

Dear Dr Gusev,

I am pleased to inform you that your manuscript entitled "The quasi-low temperature behaviour of specific heat" is now accepted for publication in Royal Society Open Science.

on behalf of Dr Robert Young (Associate Editor) and Miles Padgett (Subject Editor)
openscience@royalsociety.org

Reviewer comments to Author:
Reviewer: 2

Comments to the Author(s)
See attached file

Appendix A

The author has put forward arguments to suggest that the traditional theory of the heat capacity of solids contains inconsistencies and leads to an erroneous T^3 temperature dependence at low temperatures. In terms of an unconventional field description the author suggests that the correct power law is T^4 rather than T^3 and he presents analyses of experimental data in a range of materials in support of his assertion. This challenges one of the best known and universally accepted theories in condensed matter physics that has been taught to students since the advent of quantum mechanics.

A claim of this magnitude and fundamental nature is normally expected to be backed up by a range of compelling arguments and empirical findings. For the reasons given below this level of proof has not been reached. Despite this, however, the proposal is potentially so important that it should be made available to the scientific community in some form for deeper consideration.

The manuscript puts forward many thought provoking ideas and is of very considerable interest whether or not the principle claim is strongly supported by the evidence presented. I recommend that this work in some form be considered for publication provided that the following points are satisfactorily addressed.

1. Since the manuscript aims to convince the mainstream in condensed matter physics, the arguments for the breakdown of the conventional theory for the free energy of elementary excitations must be presented not only in terms of a new and unfamiliar language, but also in terms of the established language that is currently taught to students of physics. What precisely goes wrong with the current statistical mechanical theory of vibrational modes of a periodic lattice? It is not sufficient to simply state in general terms that the standard theory does not adequately take into account the effects of topology, i.e., the existence of finite boundaries. This is especially important given that the characteristic wavelengths of the vibrational modes that are relevant in the analyses of experimental data presented are much smaller than the characteristic sample dimensions.
2. In the conventional description, the T^3 law for the heat capacity can be understood intuitively in terms of the dynamical exponent, z , of the elementary excitations ($\omega_q \propto q^z$) and the dimension, d , of phase space available for such excitations, i.e., the heat capacity in leading order in temperature varies as $T^{d/z}$. For phonons in three dimensions, $z=1$ and $d=3$, so that the heat capacity is expected to vary as T^3 . Within this language that is widely understood, how can one justify intuitively the emergence of an extra factor of T in the author's new T^4 law? If the goal is to convince the mainstream reader then an argument entirely based on unfamiliar abstract formalism is insufficient.

3. The heat capacity data analysed by the author can be fitted to a sum of a T^3 and a T^4 term at sufficiently low T (a sum of a T^3 term and higher order terms with odd powers of T has also been considered). Importantly, the coefficient of the T^3 term is found to be finite in essentially all cases. At first sight, the latter is consistent with the conventional theory of solids, but is attributed by

the author to a contribution from the surface of the sample. This is a major claim that must be supported in a convincing manner. Minimally, the author should show that the observed T^3 component is of the form and order of magnitude expected for a surface contribution. It seems more likely to this reviewer that the order of magnitude of the coefficient of the T^3 term will be found to be consistent with the traditional theory of elementary excitations that depends essentially on the independently measureable velocity of sound. The author needs to show convincingly that this is not the case and that the origin of the T^3 is indeed unconventional.

4. The author emphasises that plots of heat capacity divided by T^3 show an initial upturn, starting from a finite $T \rightarrow 0$ intercept, followed necessarily by a peak, known as the “phonon peak”, vs. temperature. This is indeed inconsistent with the prediction of the Debye model in its conventional form. However, the issue is whether it is inconsistent with the more general, though still conventional, statistical mechanical theory of elementary excitations, if the experimentally observed dispersion relations for lattice vibrations are included. For example, we expect the correction to the linear dispersion for phonons to be negative and this by itself would tend to produce a “phonon peak”. It is possible that the “phonon peak” calculated in this way would be far too weak to account for the observed anomaly. In this case there would indeed be a problem with our conventional description that requires further investigation and the consideration of potentially new ideas.

This reviewer is intrigued by the author’s originality and many thoughtful comments and believes that there is a form of the manuscript (taking into account the above comments) that should be made available for the consideration of the more thoughtful and open-minded members of the wider scientific community.

Appendix B

Dear Dr Young,
in response to the Reviewers' Comments I substantially revised the paper. I removed four references [61-64] and the accompanying text as unessential and added 35 new ones that are used in Discussion. The manuscript has enlarged from 32 to 43 pages.

I have tried to keep changes to the main text minimal, because the Comments were mostly addressed to the criticism of other theories dealt in Discussion. Thus, only one equation was added to Sect. 1, Eq. 2, with no further changes. However, Discussion was completely redone and expanded. Now it is preceded by Conclusion, where some clarifying comments about the proposed theory are given.

Discussion now has seven subsections. Two of them contain the extended critique of the Debye theory and the Born-von Karman theory. I hope these two subsections and subsection F fully replies to the Reviewers' Comments 1 and 2. Comment 2 is not explicitly addressed but it is not a theory, but rather a common convention used for qualitative analysis. I hope Discussion explains now why this common convention, in this or other forms, may not be correct.

I have to admit that this is hardly possible to give the full explanations in one paper, which was meant to be a special case study. As can be seen from the paper and Discussion, the presented field theory of specific heat itself is only one application of the more general formalism of the thermodynamics of media, which is still in progress. I plan to communicate this progress in further publications.

In regards to Comment 3, I removed any comments about the finite-size effects in specific heat, because they are not relevant to the main content, and because a separate work on this subject is in preparation. The rest of this reply is in subsection F.

In regards to Comment 4, I think that the right theory should have predicted the 'phonon peak' instead of describing it by additional modifications. In fact, I had no idea about such phenomenon and the diverse experimental and theoretical studies of the 'boson peak' before it had emerged from my mathematical derivations. This clear prediction is an indication that the proposed axiomatic approach is working. However, it is possible that a modified phonon theory could describe it, post factum.

Best regards, Yuri Gusev.

Appendix C

Referee report

Journal: Royal Society Open Science
Article ID: RSOS-171285.R1
Title: The quasi-low temperature behaviour of specific heat
Authors: Yuri V. Gusev

Report

Earlier theoretical work by the author concerning a field theory is applied to describe the behaviour of the specific heat capacity C of crystalline and glassy solids in the so-called quasi-low temperature region. The field theory uses dimensionless thermal variables, based on the interatomic distance and the sound velocities, and provides a T^4 -behaviour of specific heat over the quasi-low temperature region, which is localized below a characteristic temperature T_0 . T_0 depends on the physical characteristics of the material and roughly corresponds to the peak temperature of the hump observed in a C/T^3 representation for both crystals and glasses. In the latter solids, it is also identified as the temperature of the calorimetric Boson peak (BP) arising from the same excess low-energy vibrations causing the BP observed in Raman spectra below 100 cm^{-1} . The analysis includes assessment of the quartic power law for the specific heat over the quasi-low temperature region in some crystals and glasses, this observation leading the author to claim the universality of this behaviour for all the solids.

This is an interesting theoretical approach on a complex argument such as the deviations of the low-temperature thermal properties of solids from the Debye predictions, and deserves publication. But the paper first needs major revision.

The English text is sometimes obscure and some revision in several places is required. Also, there are a number of typos in the text: e.g., page 10-line 3, “axes” should be “axs”; page 12-line 43: “lead” should be “leads”, etc.

The author is also kindly encouraged to improve the quality of Figures 1-12 drawing the axes on the top and the right, also including the title of plotted functions on y-axes (not only on captions).

BASIC THEME.

1. The use of definition (1) assumes the constancy of the ratio between the sound velocity V_s and the average interatomic distance a : usually in crystals, but also

in glasses, the sound velocity decreases with increasing temperature, while the thermal expansion increases. This means that the ratio V_s/a will hardly remain constant. This hypothesis appears to be quite speculative, unless the variation of this ratio over the quasi-low temperature range considered is so small than it can be neglected. Please motivate this conjecture more convincingly.

2. Figures should be reported and discussed in progressive order. At page 10, Figure 8 is referred before Figure 4 and at page 11, Figure 7 is referred before Figures 5 and 6. The author is kindly encouraged to present his description more neatly.
3. At page 11, lines 9-10, it is reported that **“this hump was dubbed in the glass matter literature “ the Boson peak” [24]...”**

The hump in C/T^3 observed in crystalline solids is probably due to some van Hove's singularity in the true lattice spectrum which, of course, is quite different from the Debye spectrum which should hold near $\nu=0$, when the material behaves as an elastic continuum. In principle, it does not correspond to the Raman or calorimetric Boson peak, observed in glasses. At present, there is a wide debate about this argument: the BP observed in glasses could be due to a van Hove's singularity, also present in the corresponding crystal, which is shifted to lower frequency/temperature by the softening of the structure (see your Ref. 85, Chumakov et al, PRL112,025502 (2014)). Alternatively, the BP could arise from localized soft vibrations of glassy units, missing in the crystal (e.g., see Carini et al, PRL 111, 245502 (2013)). The latter paper also reports the temperature behaviour of specific heat between 1.5 and 30 K (the quasi-low temperature region) in crystalline trigonal B_2O_3 , showing that it follows strictly a T^3 -behaviour in very close connection with Debye predictions. The author is kindly encouraged to comment on this in the context of his interesting proposal and also to discuss more exhaustively the debate about the possible origin of the hump in C/T^3 in crystals and glasses.

4. page 18, the sentence is not clear because the comparison between C/T^3 of glassy and crystalline forms has been performed only for PdCuP compound. Quite differently, only the heat capacity of crystalline Cu and Zn were analysed, simply because these elemental metals do not exist in glassy form.
5. page 19, lines 55-56, the first authors who discovered the collapse in a single master curve of C/C_{peak} vs T/T_{peak} for a number of glasses were Liu and Lohneisen, PRB 48, 13486 (1993).
6. page 21, authors of Ref. 45 report an experimental investigation of terahertz time-domain spectroscopy, low-frequency Raman scattering, and Brillouin light scattering on glassy glucose stating that the BP is an universal feature of

“glassy” condensed matter, and not of all the condensed matter, as reported at page 21, line 51!

RECOMMENDATION. The author should be invited to resubmit, after making the revisions suggested.

Appendix D

Reply to Comment 3 of Referee 2.

An apparent example of the third power law of specific heat in experimental data [1] was suggested by an anonymous referee. The diboron trioxide B_2O_3 was fused and placed under pressure 4 GPa. After the quenching (rapid cooling) from 1280°C to the room temperature the material was verified by the x-ray diffraction as a polycrystalline sample of the trigonal crystalline α - B_2O_3 . The measurements of Ref. [1] report, in the graphic form of C_m/T^3 vs T in Fig. 2(b), an almost perfect cubic power law - a straight horizontal line. Unfortunately, the authors did not publish the data used to plot this graph. Without published data, this experimental fact cannot be confirmed and studied.

It is difficult to make a good comparison with other measurements on the crystalline diboron trioxide, because the only data on the crystalline B_2O_3 specific heat in the literature is Ref. [3]. Ref. [1] refers to the earlier work [2] and claims using its data in Fig. 2(a). Regretfully, Ref. 15 of [1] is wrong, the cited work [2] was authored by K. K. Kelley, not by authors listed in Ref. 15. Furthermore, Ref. [2] presents data only above 51 K, that could not be useful for the purpose of [1].

However, the listed in Ref. 15 of [1] authors published the work [3] (still used in reference books), whose dataset spans temperatures from 18 to 297 K. Apparently, these data were used in Fig. 2(a) of [1]. The data of [3] are too noisy below 40 K to perform a statistical analysis. Nevertheless, the graph C_m/T^3 vs T of these data, Fig. 1, displays the right descending branch and an apparently convex top in the range 18-30 K, where the crystalline B_2O_3 characteristic temperature T_0 could possibly be. In Fig. 1 we used the units used in [1] (per gram instead of per mole) for easier comparison. One can see that the crystalline B_2O_3 specific heat seems to exhibit the usual 'anomalous' behaviour, characteristic of the T^4 power law, in the QLT regime. The authors of [1] did plot the data of [3] (by 'stars') in Fig. 2(a) because on the log-log scale used in that figure (we insist that this kind of plot is misleading and useless for quantitative analysis) these data seem to continue the T^3 line of the data, which they obtained for the quenched crystalline B_2O_3 . However, they did not use the data of [3] in Fig 2(b), C_m/T^3 vs T , perhaps, because in this kind of graph these data would not support the claimed T^3 behaviour.

The highest value of C_m/T^3 with data from [3] is $0.81 \mu J/(g \cdot K^4)$. This means that if the data plotted in Fig. 1 were plotted together with other data in Fig. 2(b), the corresponding graph would indeed join the curve for the polycrystalline B_2O_3 in Fig. 2(b), which is located

FIG. 1: C_m/T^3 vs T for crystalline B_2O_3 [3] (in units of $10^{-6} J K^{-4} g^{-1}$)

at around this height. However, it is apparent that this curve in Fig. 2(b) shows somewhat *higher* values for higher temperature, which is contradictory to the Debye cubic law. We think, that the true shape of the graph could be similar to Fig. 1, however, as is well known and clear from the data of [3] and the figure from [1], the height of the 'anomalous' hump of the graph C_m/T^3 vs T for crystalline matter is *much lower* and its peak's temperature is higher than these properties for glassy matter. Therefore, when plotted together with glassy matter graphs, the 'boson peak' of the crystalline B_2O_3 is almost distinguishable. This 'boson peak' could still be detectable by the tools of statistical analysis, if the experimental data were available.

There are two other similar experimental studies of densified glasses, germanium dioxide GeO_2 [4] and vitreous silica SiO_2 [5], by the same experimental team. Unfortunately, both studies lack the measurements for a quenched polycrystalline form of the studied compounds. We hope this experimental group would complete its work by producing such samples and measuring the specific heat of the quenched under pressure polycrystals in order to verify and study the cubic power law reported in [1].

These future measurements with tetragonal crystalline germanium dioxide could be com-

pared with the only data source, Ref. [6], which gives the specific heat of GeO_2 for $T > 16K$. These data allow us to find the quasi-low temperature threshold at $T_0 \approx 55$ K, see Fig. 2. The temperature of the 'boson peak' for the glassy GeO_2 can be estimated from Fig. 1

FIG. 2: C_m/T^3 vs T for crystalline GeO_2 [6] (in units of $10^{-5} J K^{-4} mol^{-1}$)

(b) of [4] as 8 K the normal glass and higher for densified ones. Therefore, the data [6] lie outside the region of interest of Ref. [4]. Nevertheless, the data of [6] show that properties of glassy matter are qualitatively similar to the ones of crystalline matter. The remarkable fact about the data of [6] is a very high temperature of the QLT threshold, 55 K, the only material with a higher T_0 we know of is diamond.

We could not find published data of the specific heat of crystalline silica (α or β forms of quartz) for the low temperature range considered here (high temperature data, $T > 100$ K can be found in [7], p.1755). Some 'smoothed' data taken from other references are published in [8]. The specific heat data reported in Fig. 4 of [10] and Fig. 3 (as a part of the combined plot) of [8] were obtained by E.F. Westrum Jr. and apparently published in his Cornell University thesis and [9], the references we could not obtain. Nevertheless, the C_m/T^3 vs T graph of the data from [8] has a shape characteristic to Fig. 2, with the estimated $T_0 \approx 22$ K.

Fig. 1(a) of Ref. [5] contains the graph C_m/T^3 vs T for crystalline silica, but it does not

cite the original source of data and refers instead to [10], which contains *no data* of specific heat. This fact gives grounds for the conjecture, that the authors of [5] did not plot the corresponding curve, but copied it from the figures of [10]. Indeed, this curve in Fig. 1(a) is plotted as *a line* in contrast to other graphs plotted by points.

Summarising, the authors of experimental studies [1, 4, 5] believed in the Debye cubic law, thus, they did not check properly the thermal properties of the studied compounds in crystalline form. If they would not follow the mirage of the Debye model, the conclusions of their works could be different. It could be beneficial for the physics community to see the datasets of [1, 4, 5] released into public domain. New low temperature measurements for crystalline B_2O_3 , SiO_2 and GeO_2 should be done to improve the reference books.

-
- [1] Carini GJr, Carini G, D'Angelo G, Tripodo G, Di Marco G, Vasi C, and Gilioli E. 2013 Influence of packing on low energy vibrations of densified glasses. *Phys. Rev. Lett.* **111**, 245502. (doi:10.1103/PhysRevLett.111.245502).
 - [2] Kelley KK. 1941 The specific heats at low temperatures of crystalline boric oxide, boron carbide and silicon carbide. *J. Am. Chem. Soc.* **63** (4), 1137-1139. (doi:10.1021/ja01849a072).
 - [3] Kerr EC, Hersh HN, Johnston HL. 1950 Low temperature heat capacities of inorganic solids. II. The heat capacity of crystalline boric oxide from 17 to 300 K. *J. Amer. Chem. Soc.* **72** (10), 4738-4740. (doi:10.1021/ja01166a108).
 - [4] Orsingher L, Fontana A, Gilioli E, Carini GJr, Carini G, Tripodo G, Unruh T, and Buchenau U. 2010 Vibrational dynamics of permanently densified GeO_2 glasses: Densification-induced changes in the boson peak. *J. Chem. Phys.* **132**, 124508. (doi:10.1063/1.3360039).
 - [5] Carini GJr, Carini G, Cosio D, D'Angelo G and Rossi F. 2016 Low temperature heat capacity of permanently densified SiO_2 glasses. *Philos. Mag.* **96** (7-9), 761-773. (doi:10.1080/14786435.2015.1122845).
 - [6] Counsell JF and Martin JF, The entropy of tetragonal germanium dioxide. 1967 *J. Chem. Soc. A*, part 1, **0**, 560-561. (doi:10.1039/J19670000560).
 - [7] Chase MW (ed.). 1998 *NIST-JANAF Thermochemical Tables*. 4th ed. Woodbury, NW: American Institute of Physics.
 - [8] Barron THK, Collins JF, Smith TW and White GK. 1982 Thermal expansion, Gruneisen

functions and static lattice properties of quartz. *J. Phys. C: Solid State Phys.* 15 (20), 4311-4326. (doi:doi.org/10.1088/0022-3719/15/20/016).

[9] Westrum EF Jr. 1956 In *Proceedings of the Fourth International Congress on Glass, Paris, 1956*. Chaix, Paris, p. 396.

[10] Zeller RC and Pohl RO. 1971 Thermal conductivity and specific heat of noncrystalline solids. *Phys. Rev. B* 4 (6), 2029-2041. (doi:10.1103/PhysRevB.4.2029).

Appendix E

Second Referee report

Journal: Royal Society Open Science
Article ID: RSOS-171285.R2
Title: The quasi-low temperature behaviour of specific heat
Authors: Yuri V. Gusev

Second Report

The first draft of this paper required significant revision, as I emphasized in my constructive criticism. As far as I can see, most of the points have been clarified in the revised text in a satisfactory way and the present version of the manuscript is now written with an acceptable discussion. This is an interesting theoretical approach concerning the low-temperature thermal properties of solids and could deserve publication, but the paper first needs a further important revision.

The English text continues to be sometimes obscure and there are typos in the text: e.g., page 6-first line, “vitrious” should be “vitreous”; page 6-fourth line, “It fact” should be “In fact”; page 19-line 27: “behaviour” should be “behaviour”, etc.

BASIC THEME.

1. First of all, I believe that there is no sense to include the extended comment by the author as a separate appendix entitled “Reply to Comment 3 of Referee 2”. After the requested and necessary revision, the author is strongly encouraged to include his comments about experimental observations of T^3 -behaviour in some crystals in Section V-Discussion, subsection E “The cubic law in the specific heat experimental data”.
2. I have a serious criticism on the analysis carried out on the results by Carini et al, PRL111, 245502 (2013), because it mainly highlights some meaningless details:
 - (i) it is well known that the T^3 -behaviour provided by the Debye theory is usually expected for temperatures $T \leq \theta_D/50$, in order to take the upper limit θ_D/T of the integral defining the specific heat, as large as possible (approximating infinite). This means that the observation of T^3 -behaviour for c- B_2O_3 (trigonal, $\theta_D=621$ K), c- SiO_2 (quartz, $\theta_D=550$ K) and c- GeO_2 (hexagonal, $\theta_D=365$ K and tetragonal, $\theta_D=476$ K) is

expected below about 11-12 K. Most θ_D are calorimetric, except that of quartz which is elastic, i.e. evaluated by the sound velocities. This means that for all these crystalline oxides a T^3 -behaviour is expected at temperatures below about 10 K, as experimentally observed.

- (ii) The T^3 -behaviour observed by Carini et al is not apparent, but real! Those authors emphasize correctly in caption of Fig. 2(a), “The dotted line represents a fit to the data between 2 and 8 K of c-B₂O₃, which gives a $T^{3.04}$ behavior”. As also reported in the same paper, the high temperature data of c-B₂O₃ by Kerr et al (author names correct, but journal citation wrong) have been included in the same figure, just to remark that the two different sets of experimental data are in close agreement. Surely, they were not used to prove a T^3 -behaviour over a so large interval of temperatures, because this would have been meaningless (see above). Probably, this is the only reason why Kerr's data have not been reported in Fig. 2b. T^3 -behaviours can be also clearly observed below about 12 K in tetragonal GeO₂, see the paper by Westrum and coworkers, JNCS 315, 20 (2003), and also in the hexagonal polymorph below 10 K, even if the data are really very noisy in the latter case. The same paper includes also low temperature specific heat data of c-B₂O₃ (trigonal), by Kerr et al., and of different polymorphs of c-SiO₂, showing that a T^3 -behaviour can be observed below 5 K in quartz.
- (iii) Concerning the availability of specific heat data to be used for a statistical analysis of the observed behaviors, I like to inform the author that, from many years, a number of application software are available, also on Internet and free of charge, which allow to take experimental data directly from the figures with very high accuracy; as an example, OriginPro 8 is one of these even if it is not free of charge. This means that it is not necessary, today, to have numeric values of experimental data reported on the manuscript.

Concluding, I believe that the present comments by the author about the experimentally observed T^3 -behaviour on c-B₂O₃ are quite specious and not objective in an attempt to support the possible existence of the quasi-low temperature region. They are not convincing and, above all, do not clarify the question concerning his proposal (the quasi-low temperature region) and the experimental observations in crystalline solids (including also c-GeO₂ and to a less extent c-SiO₂), which appear to follow a T^3 -behaviour for $T \leq \theta_D/50$. The author is again requested to discuss this argument more exhaustively, avoiding sterile and sometimes groundless polemics.

RECOMMENDATION. The author should be invited to resubmit, after making the suggested change.

Appendix F

I thank Referee 2 for indicating the regretful misprints, introduced by numerous revisions of the manuscript. Apparently Referee 2 does not have new major comments on the manuscript itself.

The report of Referee 2 is devoted not to my paper, but to my reply to his previous comments. One of the comments prompted me to write in TeX a larger letter, entitled "Reply to Comment 3 of Referee 2", which included the analysis of several publications and a separate list of references. As I wrote before, the paper, Carini et al, PRL111, 245502 (2013), suggested by Referee 2 in the previous report, does not have experimental data, therefore, I do not think it is possible to do rigorous scientific analysis of the results of that paper. Therefore I do not discuss this paper in my manuscript.

Let me make this clear, I wrote a long letter to the referee, not an appendix to the paper. This reply was submitted to RSOS as a separate PDF file, and no TeX file was submitted. The computer system of RSOS joined two files, my manuscript and the letter to Referee 2, which is not a part of the paper, into one PDF file. This may be confusing for referees. Perhaps, the RSOS management should address this issue in future. However, my reply is indeed incorporated into the preprint version of the paper on arxiv.org as an appendix. But that preprint is not a publication and it was not submitted to the journal. Furthermore, I am willing to remove the appendix and resubmit the arxiv preprint in a final version, if the paper is accepted by the journal.

As a personal remark to Comment 2, which is not related to my paper, I add that the claimed behaviour of the cubic power of temperature must be proven by specifying: 1) what was analysed, raw data or the derived quantity, i.e. the log-log function, 2) the number and the range of data points that were statistically analysed, 3) standard deviations, 4) the values of statistical significance. It is especially important to understand that the use of log-log plots for scientific analysis is misleading, as it caused false conclusions in different scientific fields.

I would also like to reply to Comment 3 on my concern about placing the experimental data into public domain. The journal I submitted the manuscript to, Royal Society Open Science, has an official policy of requesting authors to put all data sets into the designated depository Dryad. Thus, Royal Society also supports this idea, which makes science more transparent and meaningful. In regards to the software that could extract data from the figures, I do not think it can replace raw data. First, plotting graphs introduces new uncertainties in addition to experimental ones. Extracting data from the figures adds more uncertainties. But the worst thing is that most publications do not display raw data, they plot some derived, often complicated, functions based on the data. Sometimes, this is done on log-log or semi-log plots, then, it is really impossible to get any usable data, even by special software.

Appendix G

Third Referee report

Journal: Royal Society Open Science
Article ID: RSOS-171285.R3
Title: The quasi-low temperature behaviour of specific heat
Authors: Yuri V. Gusev

Third Report

The explanation given by the author about his extended reply to Comment 3 of Referee 2, i.e. it was to be considered just as his personal interpretation of the results reported in various papers by Carini and colleagues and not as an appendix of the manuscript, is convincing. Now, I believe that the present version of the paper is acceptable for publication.

I like to stress “**his personal interpretation**” because I do not agree with the author about the essential availability of raw experimental data to carry out a rigorous scientific analysis. The experimental data points usually reported in the figures of most papers concerning the specific heat capacity of condensed systems are more than adequate for a rigorous analysis, mostly when reported in linear scales. As emphasized in my previous report, there are a number of software programs which allow to extract experimental data directly from the figures with very high accuracy and really negligible uncertainties, mainly when the data are reported in linear scales. As a very limited example, this is the case of the papers by Carini et al (Fig. 2b of PRL111, 245502 (2013) and Fig. 1b of Phil Mag 96, 761 (2016)) and by Richet, de Ligny and Westrum (Fig.1 of JNCS 315, 20 (2003)), where the temperature behaviours of C/T^3 vs T are reported in linear scale. The data points reported in these figures are unquestionably the raw experimental data because random and systematic errors of the order of few percent are clearly visible. Surely, they are not the result of some polynomial fit used to improve and make more attractive the temperature behaviours of measured specific heat capacities.

Finally, as emphasized in my previous report, I remind to the author that the T^3 -behaviour provided by the Debye theory is usually expected for temperatures $T \leq \theta_D/50$, i.e. for temperatures lower than about 10 K in almost all the solids investigated. It has been clearly observed (in addition to the linear electronic contribution) not only in elemental crystalline metals, but also in a wide number of insulating crystals, such as c-Ge, c-KCl, c-Se, c-SiO₂ (quartz), crystalline Polyethylene (Figs. 3.2 and 3.3 of the paper by Robert Pohl, in *Low Temperature*

Specific Heat of Glasses in Amorphous Solids: Low-Temperature properties, ed. by W. A. Phillips, Springer, Berlin 1981, pp 27-52), c-GeO₂ (tetragonal, Fig. 3.17 of the same paper by Pohl and Fig.1 of the paper by Westrum), c-B₂O₃ (trigonal, Fig. 2b of the paper by Carini et al). In all these cases, the T⁴-behaviour provided by your quasi-low temperature model concerns the temperature region above few Kelvin where a peak is observed in a C/T³ representation of the specific heat capacity. The peak arises from the fact that the real frequency spectrum in the atomic lattice is very different from $g(\nu) \propto \nu^2$ assumed by Debye and can be attributed to a peak in the phonon density of states due to an unusually flat dispersion relation of the lowest transverse acoustic mode near the Brillouin-zone boundary. The only exception is crystalline polyethylene, which strictly follows the Debye behaviour without any peak, see the paper by Pohl, Fig.3.3.